# GD2-CAR T cell therapy for H3K27M-mutated diffuse midline gliomas

Robbie G. Majzner[1,2,3,13], Sneha Ramakrishna[1,2,13], Kristen W. Yeom[4], Shabnum Patel[1], Harshini Chinnasamy[1], Liora M. Schultz[1,2], Rebecca M. Richards[1,2], Li Jiang[5], Valentin Barsan[1,2], Rebecca Mancusi[6], Anna C. Geraghty[6], Zinaida Good[1,3,7], Aaron Y. Mochizuki[6], Shawn M. Gillespie[6], Angus Martin Shaw Toland[8], Jasia Mahdi[6], Agnes Reschke[1,2], Esther H. Nie[6], Isabelle J. Chau[6], Maria Caterina Rotiroti[2], Christopher W. Mount[6], Christina Baggott[1], Sharon Mavroukakis[1], Emily Egeler[1], Jennifer Moon[1], Courtney Erickson[1], Sean Green[2], Michael Kunicki[1,2], Michelle Fujimoto[1,2], Zach Ehlinger[2], Warren Reynolds[2], Sreevidya Kurra[2], Katherine E. Warren[5], Snehit Prabhu[1], Hannes Vogel[8], Lindsey Rasmussen[9], Timothy T. Cornell[9], Sonia Partap[6], Paul G. Fisher[6], Cynthia J. Campen[6], Mariella G. Filbin[5], Gerald Grant[10], Bita Sahaf[1,2], Kara L. Davis[1,2], Steven A. Feldman[1], Crystal L. Mackall[1,2,3,11,14 ✉] & Michelle Monje[1,2,6,8,10,12,14 ✉]

Diffuse intrinsic pontine glioma (DIPG) and other H3K27M-mutated diffuse midline gliomas (DMGs) are universally lethal paediatric tumours of the central nervous system[1]. We have previously shown that the disialoganglioside GD2 is highly expressed on H3K27M-mutated glioma cells and have demonstrated promising preclinical efficacy of GD2-directed chimeric antigen receptor (CAR) T cells[2], providing the rationale for a first-in-human phase I clinical trial (NCT04196413). Because CAR T cell-induced brainstem inflammation can result in obstructive hydrocephalus, increased intracranial pressure and dangerous tissue shifts, neurocritical care precautions were incorporated. Here we present the clinical experience from the first four patients with H3K27M-mutated DIPG or spinal cord DMG treated with GD2-CAR T cells at dose level 1 (1 × 10⁶ GD2-CAR T cells per kg administered intravenously). Patients who exhibited clinical benefit were eligible for subsequent GD2-CAR T cell infusions administered intracerebroventricularly[3]. Toxicity was largely related to the location of the tumour and was reversible with intensive supportive care. On-target, off-tumour toxicity was not observed. Three of four patients exhibited clinical and radiographic improvement. Pro-inflammatory cytokine levels were increased in the plasma and cerebrospinal fluid. Transcriptomic analyses of 65,598 single cells from CAR T cell products and cerebrospinal fluid elucidate heterogeneity in response between participants and administration routes. These early results underscore the promise of this therapeutic approach for patients with H3K27M-mutated DIPG or spinal cord DMG.

This phase I dose-escalation trial of autologous GD2-CAR T cells (containing a GD2 binding domain, a 4-1BB co-stimulatory domain and a CD3z signalling domain) in children and young adults with pontine and spinal cord DMG characterized by a K27M mutation in genes encoding histone H3 (H3K27M) was designed with the primary objectives of assessing feasibility of manufacturing, safety and tolerability, and identifying the maximally tolerated dose or recommended phase II dose (Fig. 1a, b). Assessment of clinical activity was a secondary objective and identifying biomarkers of response was an exploratory objective. We anticipated the development of neurological symptoms related to CAR T cell-mediated inflammation in sites of central nervous system (CNS) disease[2], which we have termed tumour inflammation-associated neurotoxicity (TIAN). To mitigate risks associated with TIAN, we excluded patients with bulky

[1]Stanford Center for Cancer Cell Therapy, Stanford Cancer Institute, Stanford University, Stanford, CA, USA. [2]Division of Pediatric Hematology, Oncology, Stem Cell Transplantation & Regenerative Medicine, Department of Pediatrics, Stanford University, Stanford, CA, USA. [3]Parker Institute for Cancer Immunotherapy, San Francisco, CA, USA. [4]Division of Neuroradiology, Department of Radiology, Stanford University, Stanford, CA, USA. [5]Division of Pediatric Neuro-Oncology, Dana Farber Cancer Institute, Boston, MA, USA. [6]Department of Neurology and Neurological Sciences, Stanford University, Stanford, CA, USA. [7]Department of Biomedical Data Science, Stanford University, Stanford, CA, USA. [8]Department of Pathology, Stanford University, Stanford, CA, USA. [9]Division of Critical Care Medicine, Department of Pediatrics, Stanford University, Stanford, CA, USA. [10]Department of Neurosurgery, Stanford University, Stanford, CA, USA. [11]Division of Stem Cell Transplantation and Cell Therapy, Department of Medicine, Stanford University, Stanford, CA, USA. [12]Howard Hughes Medical Institute, Stanford University, Stanford, CA, USA. [13]These authors contributed equally: Robbie G. Majzner, Sneha Ramakrishna. [14]These authors jointly supervised this work: Crystal L. Mackall, Michelle Monje. ✉e-mail: cmackall@stanford.edu; mmonje@stanford.edu

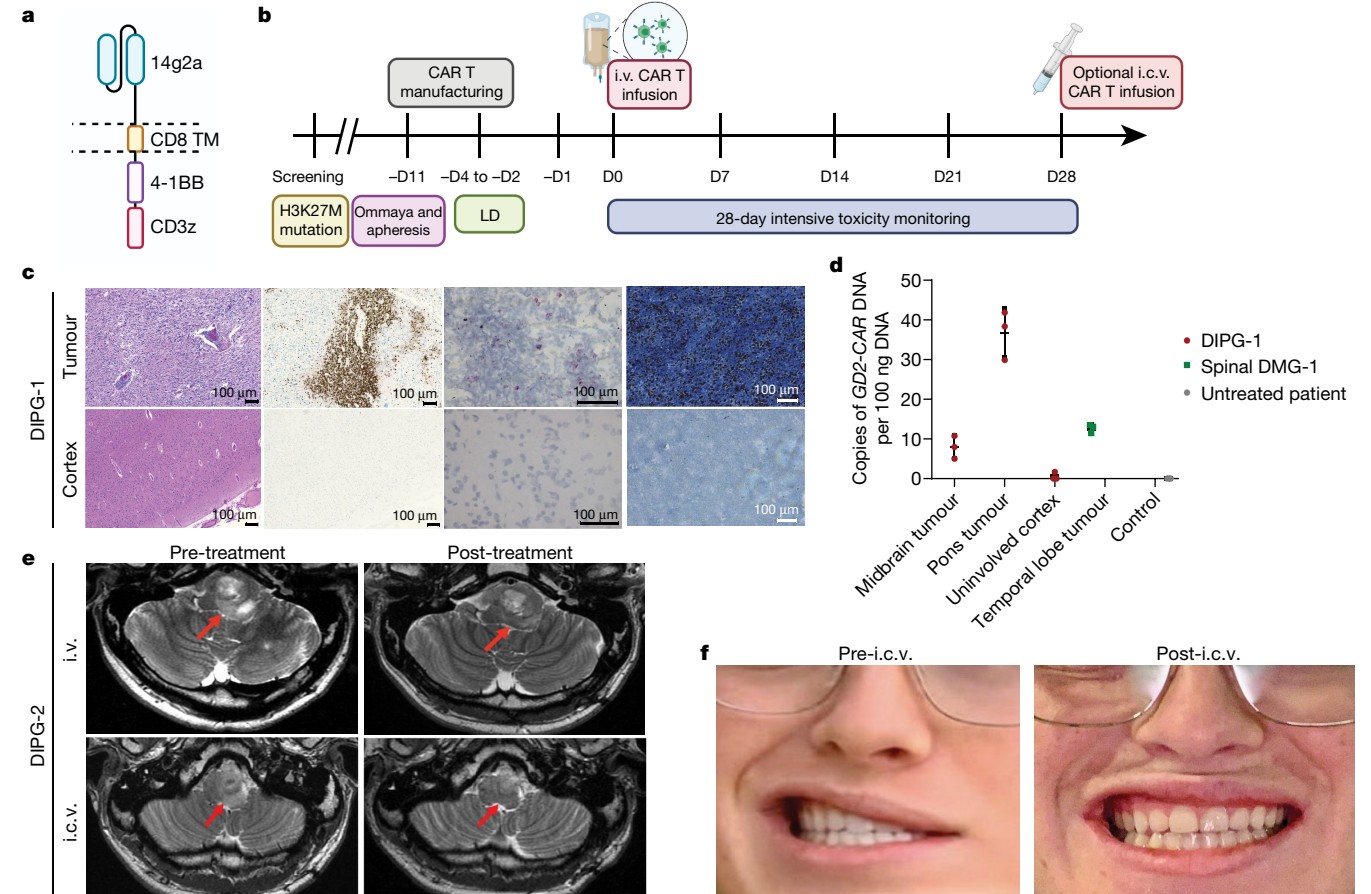

**Fig. 1 | Trial design and patients 1 and 2 with DIPG. a**, GD2–4-1BB–CD3Z CAR schematic. TM, transmembrane domain. **b**, Outline of clinical trial design. D0, day 0; LD, lymphodepleting chemotherapy. **c**, Post-mortem examination of participant 1 with DIPG (DIPG-1). From left to right: haemotoxylin and eosin staining; CD3 immunohistochemistry (brown); *GD2-CAR* mRNA puncta (pink; haematoxylin counterstain for all cells in blue); and GD2 antigen (blue) immunohistochemistry (H3K27M⁺ nuclei in brown). **d**, qPCR for *GD2-CAR* DNA from autopsy samples from DIPG-1 exhibit the presence of GD2-CAR in tumour-involved midbrain and pons, but not her uninvolved cortex or brain tissue from an untreated individual (control). Resected temporal lobe tumour

from patient 1 with spinal DMG (spinal DMG-1) following i.v. infusion reveals the presence of *GD2-CAR* DNA. Data represent mean ± s.e.m., *n* = 3 technical replicates for each sample. **e**, MRI scan (axial T2, shown at the level of the lower pons) of participant 2 with DIPG (DIPG-2) before and 4 weeks after i.v. infusion (top row). An MRI scan before and 2 weeks following i.c.v. infusion is also shown (bottom row). A reduction in the tumour size is observed after each infusion (red arrows). **f**, Photographs of DIPG-2 demonstrate significant improvement in left facial strength 2 weeks after i.c.v. infusion. Photographs were obtained and published with informed consent. Schematics were created with BioRender. com.

---

thalamic or cerebellar tumours, required placement of an Ommaya reservoir in patients with DIPG to monitor intracranial pressure (ICP), and instituted a TIAN toxicity management algorithm incorporating the removal of cerebrospinal fluid (CSF) via Ommaya, hypertonic saline, anti-cytokine agents and corticosteroids.

## Dose level 1

Beginning June 2020, four participants were enrolled on dose level 1 (DL1; 1 × 10⁶ GD2-CAR T cells per kg administered intravenously; three participants with DIPG and one participant with spinal cord DMG; 5–25 years of age; one male and three female; Supplementary Table 1); all participants were more than 6 months from completion of standard radiotherapy. GD2-CAR T cells were successfully manufactured and met release criteria for all four patients (Supplementary Table 2). During cell manufacturing, patient 1 with spinal DMG experienced rapid tumour progression and was removed from the study but was treated at DL1 on a single-patient compassionate emergency investigational new drug application (eIND). Results are reported here with a data cut-off of March 2021.

Patient 1 with DIPG was a 14-year-old girl with H3K27M⁺ DIPG, with early radiographic signs of tumour progression, right sixth nerve palsy, left facial weakness, dysarthria, bilateral dysmetria and wide-based gait at the time of treatment. On day +6 following treatment with GD2-CAR T cells intravenously (i.v.), she experienced grade 1 cytokine release syndrome (CRS; 40.1 °C fever) together with worsening cranial nerve symptoms consistent with TIAN and was treated with tocilizumab (an IL-6 antagonist) and corticosteroids (Supplementary Table 3). On day +9, she experienced an acute episode of fever, hypertension, decreased responsiveness, hemiplegia and extensor posturing. Her Ommaya was immediately accessed; her ICP was elevated at 22 mmHg, 10 ml CSF was drained, and she returned to her neurological baseline within minutes. Anakinra (an IL-1R antagonist) was started. MRI demonstrated increased pontine oedema. She remained on corticosteroids, with transient improvement in her baseline dysarthria and dysmetria. One month post-infusion, increased disease (approximately 20% enlargement) was evident on MRI, followed by tumour progression resulting in death 3 months after infusion of GD2-CAR T cells, 3 months after radiographic progression was first noted and 13 months after diagnosis (Extended Data Fig. 1a, b). Post-mortem brain examination

demonstrated substantial infiltration of lymphocytes in the tumour that is uncharacteristic for DIPG[4]; lymphocytic infiltration was not observed in areas of the normal brain (Fig 1c, Extended Data Fig. 1c). RNAscope identified cells expressing *GD2-CAR* mRNA transcript in the tumour, but not the unaffected cortex (Fig. 1c, Extended Data Fig. 1d). Similarly, the *GD2-CAR* transgene was detected by quantitative PCR (qPCR) of DNA from tumour tissue (Fig. 1d). GD2-antigen expression was substantially higher in the tumour than in the normal brain tissue (Fig. 1c, Extended Data Fig. 1e). Tumour microglial and other myeloid cell infiltration was prominent (Extended Data Fig. 1f, g).

Patient 2 with DIPG was a 21-year-old man with H3K27M[+] DIPG exhibiting early signs of clinical and radiographic progression at the time of enrolment, with left sixth nerve palsy, left facial weakness, decreased left facial sensation, trismus with difficulty opening his mouth that limited large bites of food, right lower extremity sensory loss and wide-based gait with inability to tandem walk. On day +7 following i.v. GD2-CAR T cell administration, he developed grade 2 CRS (fever to 39.4 °C and hypotension responsive to fluids) and transient worsening of baseline deficits consistent with TIAN affecting pontine function, including transiently worsened trismus limiting oral intake to eating through a straw, transient right upper extremity sensory loss and mild left hearing loss lasting 2 days. MRI demonstrated increased T2/FLAIR signal in the left trigeminal nucleus (Extended Data Fig. 2a), which controls the muscles of mastication, correlating with the symptom of trismus. He was treated with tocilizumab and anakinra, but no corticosteroids (Supplementary Table 3). By the second week, he experienced marked improvement in trismus (near-normal mouth opening and no limitations to eating), improvement in facial symmetry, resolution of baseline sensory loss and improvement in gait with the ability to tandem walk (Supplementary Table 4). By 1 month following treatment with GD2-CAR T cells, he exhibited a near-normal neurological examination, with only a residual sixth nerve palsy. MRI demonstrated areas of improved T2/FLAIR signal and mildly decreased (approximately 17% smaller) tumour volume (Fig. 1e, Extended Data Fig. 2b).

Clinical and radiographic improvement persisted until 2–3 months, when he experienced recrudescence of previous symptoms, including return of trismus, facial weakness and sensory loss, left hearing loss, increased weakness and sensory loss affecting the right side of his body, and gait instability that required wheelchair use for any distance beyond a few steps. He received a second dose of $30 \times 10^6$ GD2-CAR T cells administered intracerebroventricularly (i.c.v.) via Ommaya reservoir without lymphodepletion. He developed a fever (40 °C) within 24 h and approximately 48 h post-infusion developed acutely increased somnolence and new right third nerve palsy. His Ommaya reservoir was accessed, demonstrating elevated ICP of 34 mmHg; CSF was removed with immediate clinical improvement. Neuroimaging demonstrated obstructive hydrocephalus due to compression of the fourth ventricle by the expanded pons. The Ommaya was left accessed for continuous CSF drainage. Hypertonic saline, anakinra and systemic corticosteroids were administered (Supplementary Table 3). Hydrocephalus resolved without need for continued drainage of CSF after 2 days. Corticosteroids were discontinued after 4 days (on day +6). Two weeks post-i.c.v. infusion, his neurological examination had markedly improved, with near-normalization of facial strength (Fig. 1f) and sensation, improved trismus, hearing, right-sided weakness and dramatic improvement in gait that enabled independent walking for long distances (Supplementary Videos 1, 2, Supplementary Table 4). An MRI scan 2 weeks post-i.c.v. GD2-CAR T cell infusion demonstrated 27% reduction in tumour volume compared to his pre-i.c.v. infusion MRI (Fig. 1e, Extended Data Fig. 2b).

After the data cut-off of March 2021, he went on to receive three more i.c.v. infusions (five infusions total). Before the planned sixth infusion, he died due to an intratumoural haemorrhage in a known area of intratumoural vascular anomaly. Such intratumoural haemorrhages are relatively common in DIPG and risk increases with time from

diagnosis[5]. He survived 26 months from diagnosis, 12 months after he began to exhibit radiographic and clinical progression, and 10 months after his first GD2-CAR T cell infusion.

Patient 3 with DIPG was a 5-year-old girl with H3K27M[+] DIPG who enrolled before progression. She did not exhibit brainstem enlargement, only areas of infiltrative tumour evident as patchy abnormal T2/FLAIR signal in the left cerebral peduncle (corticospinal tract motor fibres) of the midbrain (Fig. 2a), anterior pons, medulla and left cerebellar peduncle. At baseline, she exhibited bilateral facial weakness, hypophonic voice, rightward tongue deviation, difficulty controlling oral secretions (drooling), chronic nausea, hypertonic right hemiparesis that limited use of the right arm and hand (her right hand was held in a flexed position; Fig. 2b), dysmetria and a hemiparetic, wide-based gait. On day 7 after i.v. administration of GD2-CAR T cells, she developed grade 1 CRS (fever of 40.4 °C) and transiently increased ataxia, consistent with TIAN. She was treated with tocilizumab and anakinra, but no corticosteroids (Supplementary Table 3). By 2 weeks post-infusion, her chronic nausea (present since diagnosis) had resolved, her right-sided motor function improved and she was using her right hand spontaneously in play, with improved ability to extend her fingers (Fig. 2b). By 1 month post-infusion, she was controlling oral secretions without drooling, taking bigger and stronger bites of food, speaking with a louder voice and showing improved facial strength that enabled a wide smile. These clinical improvements were accompanied by decreased T2/FLAIR MRI signal abnormality in the midbrain (Fig. 2a, c). By 6 weeks after i.v. GD2-CAR T cells, her balance, coordination and use of her right leg had improved, enabling her to ride a scooter using her right leg (Supplementary Table 4). With resolution of chronic nausea, her food intake improved; she increased from the sixth percentile to the thirtieth percentile for weight and grew 2 inches in height in the 2 months following i.v. GD2-CAR T cells.

She was re-treated with $12.9 \times 10^6$ GD2-CAR T cells i.c.v. (equivalent to $1 \times 10^6$ per kg) without lymphodepletion 3 months after her i.v. infusion, when improvement in her clinical examination had plateaued and increased cerebellar peduncle disease was evident (Fig. 2d, Extended Data Fig. 2c). She experienced only mild headache. No anti-cytokine agents or steroids were used. Her clinical response was mixed. She demonstrated additional improvement in right-sided motor function (improved right ankle dorsiflexion and wrist extension, and further increased use of right hand) within 1 week of her second GD2-CAR T cell infusion (Supplementary Table 4). By contrast, her left-sided ataxia progressively worsened consistent with increased left cerebellar peduncle disease (Fig. 2d, Extended Data Fig. 2c).

After the data cut-off of March 2021, this participant received one more i.c.v. infusion but her tumour continued to progress. She died from tumour progression 20 months after diagnosis, 4 months after she had begun to exhibit tumour progression, and 7 months after her first GD2-CAR T cell infusion.

Patient 1 with spinal DMG was a 25-year-old woman with an H3K27M[+] spinal cord DMG centred at thoracic levels T10–T12 who exhibited early signs of clinical and radiographic progression at the time of enrolment. After trial enrolment and during cell manufacturing, she experienced rapid tumour progression resulting in near-complete paraparesis, severe neuropathic pain, sensory loss below the T10 level, bladder dysfunction requiring urinary catheterization, and tumour spread to her temporal lobe. An urgent duraplasty was performed to relieve pressure in the spinal canal due to the rapidly expanding, tumour-infiltrated cord and to mitigate her severe neuropathic pain. Pain was moderately improved after duraplasty, but she still required a lidocaine drip and opiates for pain management. She no longer met criteria for GD2-CAR T cell infusion on-study due to rapid tumour progression with neurological deterioration and corticosteroid use before GD2-CAR T cell infusion and was removed from the protocol. She received GD2-CAR T cells ($1 \times 10^6$ per kg) administered i.v. on an eIND. Beginning day +6, she exhibited grade 3 CRS (fever to 39.6 °C

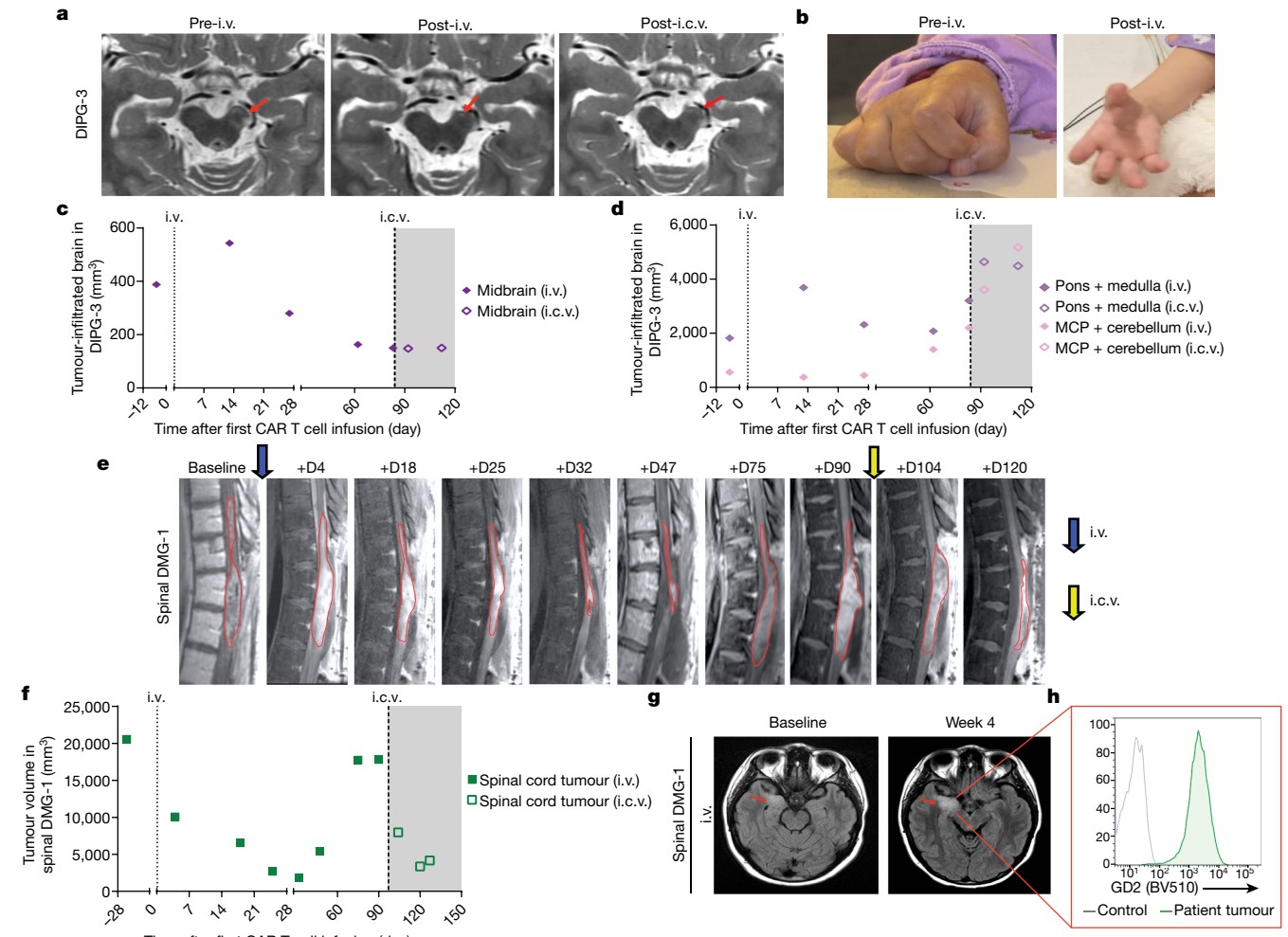

**Fig. 2 | Patient 3 with DIPG and patient 1 with spinal DMG. a**, MRI images (axial T2) of participant 3 with DIPG (DIPG-3) showed a decrease in abnormal T2 signal (tumour) in the left cerebral peduncle corticospinal tract motor fibres (red arrows) 4 weeks following i.v. infusion that remained stably improved (MRI 2 weeks following i.c.v. infusion is also shown). **b**, The right hand of DIPG-3 at baseline, which exhibited poor strength, increased tone and was held in chronic flexion. Recovery of right-hand motor function was observed by 2 weeks after i.v. treatment, with increased movement and improved tone. **c**, Midbrain (left cerebral peduncle) tumour volume change over time in DIPG-3. **d**, Tumour volume change in the pons and medulla, and the middle

cerebellar peduncles (MCP) and cerebellum over time in DIPG-3. **e**, Sagittal MRI images of patient 1 with spinal cord DMG (spinal DMG-1) show a decrease in the tumour (outlined in red) following i.v. treatment (blue arrow) and i.c.v. re-treatment (yellow arrow). **f**, Change over time in spinal cord tumour volume in spinal DMG-1. Grey shading indicates time period following i.c.v. infusion (**c**,**d**,**f**). **g**, Despite significant tumour reduction in **f**, a temporal lobe tumour (red arrow) in spinal DMG-1 did not respond (axial T2 MRI images). **h**, GD2 expression in the resected temporal lobe tumour from spinal DMG-1 was high and uniform by flow cytometry as compared to a fluorescence-minus-one control.

and hypotension requiring vasopressor support), which was treated with tocilizumab and corticosteroids for 3 days (days +8–11; Supplementary Table 3). Despite the trajectory of her rapid tumour progression before GD2-CAR T cell treatment, imaging demonstrated more than 90% reduction in volume of the spinal cord tumour by day +32 (Fig. 2e, f) accompanied by improved lower extremity motor and urinary function, resolution of neuropathic pain and discontinuation of all pain medications by 2 months post-infusion (Supplementary Table 4). By contrast, her brain metastasis did not exhibit improvement (Fig. 2g).

Both brain and spinal cord disease progressed by day +75 (Fig. 2e, f). The tumour had spread extensively throughout her brain (Extended Data Fig. 2d), and the bulkiest component in the medial temporal lobe was resected. Flow cytometry of the resected tissue demonstrated robust tumour cell GD2 expression (Fig. 2h) but few infiltrating T cells. Low levels of the *GD2-CAR* transgene were detected by qPCR in the resected tumour of the temporal lobe (Fig. 1d). With limited therapeutic options, she was re-treated with 50 × 10⁶ GD2-CAR T cells administered i.c.v. following increased lymphodepletion, given the concern for

immune rejection of the CAR T cells. The chosen dose for CAR T cells was within the range of cell doses administered i.c.v. on other brain tumour trials[6,7]. She was the first patient chronologically to receive GD2-CAR T cells i.c.v., and her course informed the i.c.v. strategy described above. Within 48 h, she developed persistent fevers (up to 39.9 °C) and grade 3 encephalopathy (grade 4 immune effector cell-associated neurotoxicity (ICANS)), associated with no MRI changes in the normal brain and diffuse slowing with triphasic waves on continuous EEG monitoring, consistent with a reversible toxic/metabolic/inflammatory encephalopathy (Extended Data Fig. 3). Given the extensive tumour invasion of brain structures including the thalami, hypothalamus, mammillary bodies, bilateral insula and bilateral frontal lobes, we were unable to distinguish between ICANS and TIAN as the primary cause for her encephalopathy. She was treated with anti-inflammatory therapy (anakinra, siltuximab (an IL-6 antagonist) and corticosteroids (i.v. and i.c.v.)) and oral dasatinib to dampen the activity of CAR T cells[8] (Supplementary Table 3). Within 4 days, the encephalopathy resolved. Steroids were weaned and discontinued by day +24. Regression was again observed

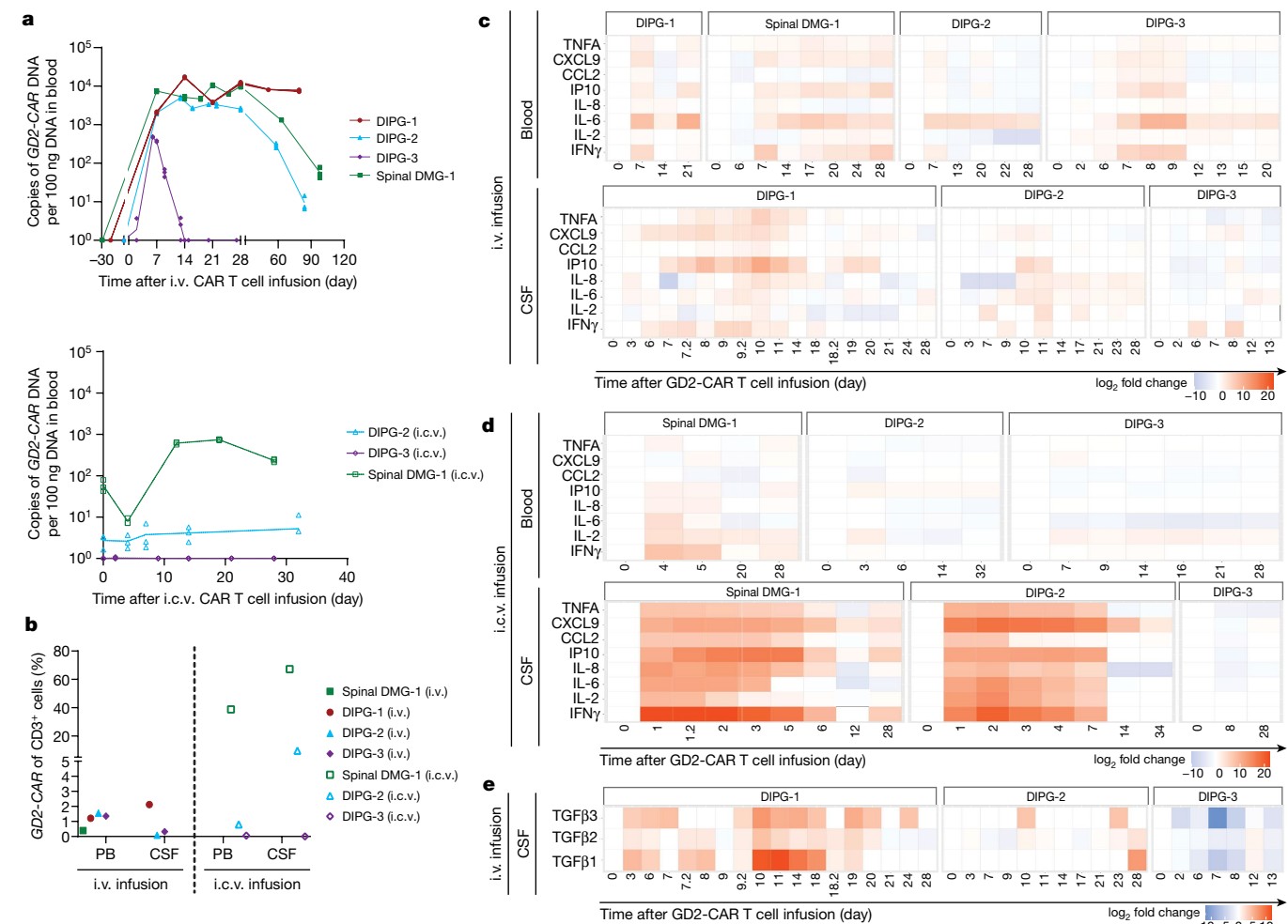

**Fig. 3 | CAR T cell kinetics and cytokine production. a**, qPCR for *GD2-CAR* DNA illustrates kinetics of CAR T cell expansion and persistence in the peripheral blood following i.v. (top) and i.c.v. (bottom) administration. Each point represents one technical replicate; *n* = 3 technical replicates per timepoint per patient, 4 patients (i.v.), 3 patients (i.c.v.). **b**, Flow cytometry of GD2-CAR demonstrated significantly higher proportion of CAR T cells in the CSF following i.c.v. GD2-CAR T cell infusion than following i.v. infusion at peak inflammation timepoints. Each dot represents one patient. *n* = 4 patients (i.v.), 3 patients (i.c.v.). **c**, **d**, Pro-inflammatory cytokines in the blood and CSF after i.v. infusion (**c**), *n* = 4 patients, and after i.c.v. infusion (**d**), *n* = 3 patients. Note that the expression of IFNγ in the CSF of spinal DMG-1 on D1–3 after i.c.v. infusion was above the upper limit of detection of the assay. **e**, Levels of the immunosuppressive cytokines TGFβ1, TGFβ2 and TGFβ3 in the CSF after i.v. infusion. *n* = 3 patients. Heatmaps in **c**–**e** were generated from Luminex multiplex cytokine analysis of patient blood plasma and CSF in technical duplicates. Average pg ml⁻¹ results are represented by log₂ fold change from D0 timepoint.

in her spinal tumour (more than 80% reduction over 3 weeks; Fig. 2e, f) as well as select areas of her brain disease (Extended Data Fig. 2d). Clinical improvement in spinal cord function was not observed after the second infusion as it was after the first (Supplementary Table 4). Both spinal and brain disease progressed further by 2 months following i.c.v. infusion and she received no further GD2-CAR T cell infusions. After the data cut-off of March 2021, she survived 20 months from diagnosis, 14 months from the beginning of tumour progression and 11 months after her first GD2-CAR T cell infusion.

## Correlative findings

Serum and CSF samples were obtained routinely and at times of clinical intervention to assess markers of inflammation and correlates of response. Patient 1 with spinal DMG had CSF samples collected following only i.c.v. administration but not following i.v. administration, and limited CSF samples were obtained from patient 3 with DIPG due to her young age.

Correlating with tumour response, serum levels of lactate dehydrogenase increased in all patients after GD2-CAR T cell treatment and

tracked with evidence of inflammation (Extended Data Fig. 4a). Using digital droplet PCR for the tumour-specific H3K27M mutation, cell-free tumour DNA was detected in CSF in two of four patients (Extended Data Fig. 4b–f). CSF cell-free tumour DNA was elevated during peak inflammation post-i.c.v. treatment, a pattern expected with tumour cell killing[9]. Additional sampling may further elucidate the kinetics of tumour cell killing in future studies.

GD2-CAR T cell expansion and persistence was monitored using *GD2-CAR* transgene qPCR and flow cytometry of cell-surface CAR expression. GD2-CAR T cell expansion in blood assessed by qPCR was similar in magnitude to expansion of highly active CAR T cells for haematological malignancies[10–12] (Fig. 3a). Flow cytometry-based GD2-CAR T cell detection in blood was limited, potentially due to decreased surface expression of GD2-CAR after activation[13]. GD2-CAR T cells were detected in CSF by flow cytometry, particularly following i.c.v. administration (Fig. 3b, Extended Data Fig. 6b).

In blood, cytokines implicated in CRS, including IL-6, were higher following i.v. than i.c.v. administration, consistent with the higher grade CRS observed after i.v. administration. In CSF, the levels of

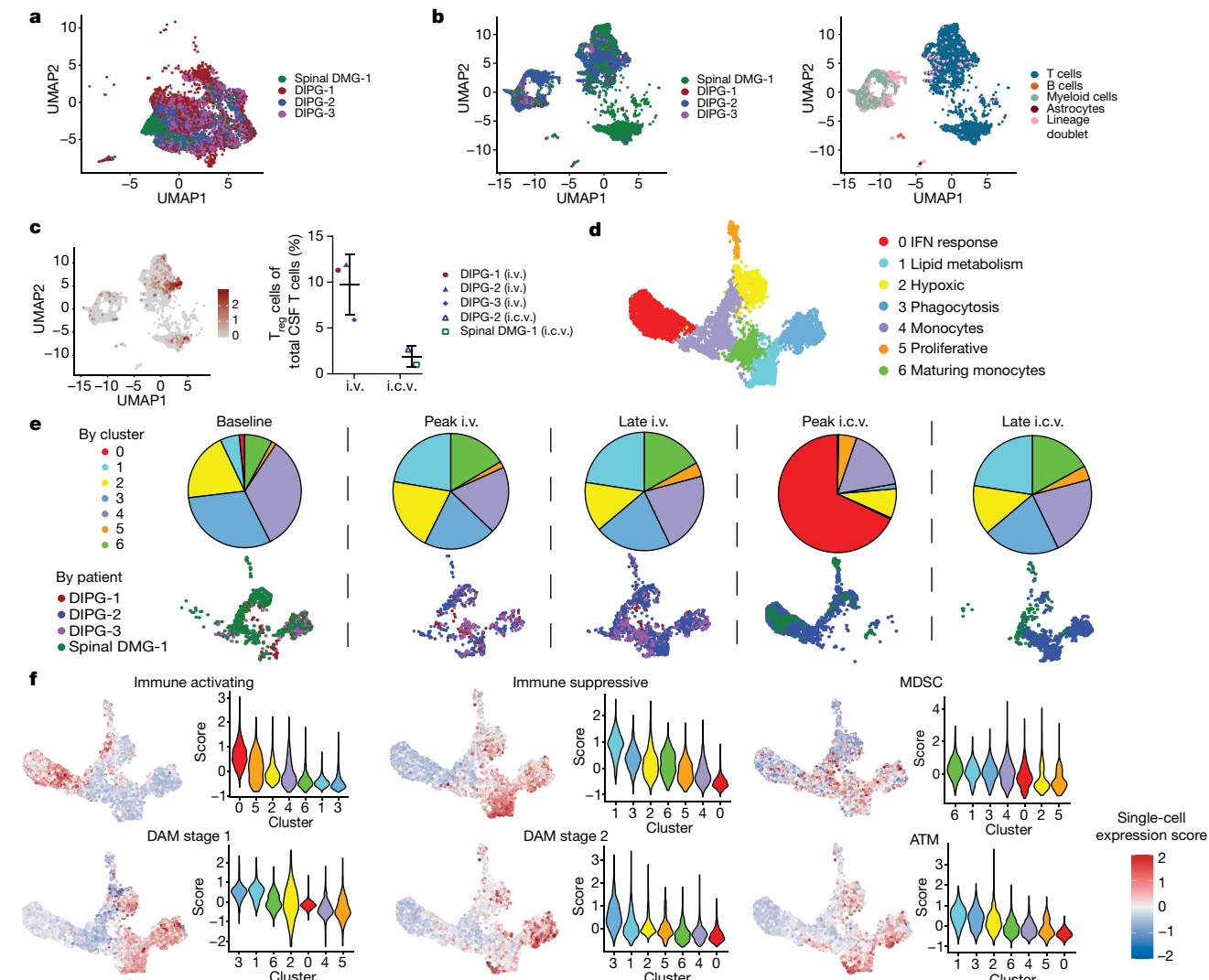

**Fig. 4 | Single-cell transcriptomic analyses identifies distinct myeloid subpopulations. a**, UMAP representation identifies cellular populations within GD2-CAR⁺ flow-sorted T cell products. $n$ = 20,000 single cells (5,000 GD2-CAR⁺ cells from each of the 4 patient products). **b**, UMAP representation identifies cellular populations within CSF samples from patients. CSF cells by patient (left) and CSF cells by cell type (right) are shown. $n$ = 25,598 single cells. **c**, UMAP highlighting FOXP3⁺ regulatory T ($T_{reg}$) cells (left). $T_{reg}$ cell population (defined by CD4, FOXP3 and CD25 expression) identified in CSF samples at peak inflammation timepoints following i.v. or i.c.v. administration (right). $n$ = 523 $T_{reg}$ cells were analysed from a total of 17,699 CSF T cells from 3 participants after i.v. and 2 participants after i.c.v. administration. With a Bayesian model-based single-cell compositional data analysis (scCODA)

framework, the log fold change was 0.6, the inclusion probability was 0.75 and the false discovery rate was less than 0.05. **d**, CSF sample single-cell RNA sequencing was filtered to isolate myeloid cells. Clustering was conducted after data integration by Harmony. $n$ = 6,497 myeloid cells. **e**, Pie charts represent myeloid cluster proportions at different timepoints. A single UMAP was generated with myeloid cells from the CSF, and then individually visually represented based on timepoint of the sample, coloured by patient. Note the alterations in the presence of myeloid clusters over time. **f**, Expression signature of 'immune activating', 'immune suppressive', 'disease-associated myeloid (DAM) stage 1', 'DAM stage 2', 'myeloid-derived suppressor cell (MDSC)' and 'axon tract-associated microglial (ATM)' cell states and associated representation within clusters based on single-cell expression scores ($Z$ score).

pro-inflammatory cytokines, including IFNγ, TNF, IL-2 and IL-6, were higher following i.c.v. administration than following i.v. administration (Fig. 3c, d, Extended Data Fig. 5a–c). Patient 1 with DIPG exhibited increased levels of immunosuppressive cytokines, including TGFβ, in the CSF (Fig. 4e, Extended Data Fig. 5d) and prominent myeloid populations in tumour tissue (Extended Data Fig. 1f, g), which may correlate with her lack of response. Cytokine analyses confirmed that anakinra (modified recombinant IL-1RA) crossed the blood–brain barrier, with detectable levels of IL-1RA found in both the blood and the CSF following i.v. administration of anakinra (Extended Data Fig. 5e).

Single-cell RNA sequencing was performed on cells isolated from CSF (Extended Data Fig. 6a) as well as flow-sorted GD2-CAR-positive and GD2-CAR-negative T cells from the manufactured products (Fig. 4a, b). We analysed 65,598 single cells: 25,598 cells from CSF and

40,000 cells from manufactured products. The cellular composition of CSF included lymphocytes, myeloid cells, rare astrocytes and B cells (Fig. 4b). We observed a population of lineage doublets, raising the possibility of phagocytosis of T cells by myeloid cells (Fig. 4b). We observed more GD2-CAR T cells in CSF at times of peak inflammation, and more CSF GD2-CAR T cells after i.c.v. versus i.v. administration for patient 2 with DIPG, the only patient in whom this comparison was possible (Extended Data Fig. 6b). By single-cell RNA sequencing, we did not identify GD2-CAR T cells in the CSF of patients 1 and 3 with DIPG. Comparing manufactured cell products for patient 1 with DIPG, who did not have a response, to patient 2 with DIPG, who exhibited radiographic and clinical benefit, revealed numerous transcriptional differences. These included baseline interferon and PD1 signalling activation pathways in patient 1 with DIPG, potentially indicating

increased tonic signalling associated with reduced GD2-CAR T cell activity[14] (Extended Data Fig. 6c, d). Comparisons of other products are shown in Extended Data Fig. 6e, f. Assessing all lymphocytes from the CSF of all participants revealed that the T cell CD4:CD8 ratios in CSF differed over time between patients (Extended Data Fig. 7a). Increased numbers of regulatory T ($T_{reg}$) cells were found in patient CSF after i.v. GD2-CAR T cell administration compared to i.c.v. administration (Fig. 4c, Extended Data Fig. 7b).

Examination of all CSF myeloid cells from all patients after each administration route revealed seven distinct clusters of cells, including monocytes, microglia and macrophages expressing various functional signatures, and proliferating myeloid cells (Fig. 4d, Extended Data Fig. 8a, Supplementary Table 5). We identified a myeloid population characterized by interferon response, present primarily at peak inflammation timepoints after i.c.v. administration (Fig. 4d, e). This cluster exhibited a distinctly immune-activating signature (Fig. 4f, Extended Data Fig. 8b), concordant with the pro-inflammatory CSF cytokine profile observed after i.c.v. administration (Fig. 3c, d). By contrast, i.v. administration and late timepoints had CSF myeloid cell subpopulations that expressed prominent phagocytosis and lipid metabolism gene programmes. These cells also exhibited a strong immune-suppressive profile, which aligned with transcriptional signatures of disease-associated microglia[15], myeloid-derived suppressor cell and axon tract-associated microglia[16] cell states (Fig. 4e, f, Extended Data Fig. 8). Comparison of the CSF myeloid cell fraction in patient 1 with DIPG to other patients at the time of peak inflammation highlighted differentially increased interleukin, chemokine and neutrophil degranulation processes (Extended Data Fig. 9a–c). Similarly, comparison of the CSF myeloid cell fraction in patient 2 with DIPG at peak inflammation following i.v. versus i.c.v. administration revealed increased myeloid activation processes following i.v. administration (Extended Data Fig. 9d).

Together, these correlative studies provide further evidence of GD2-CAR T cell activity against H3K27M+ DIPG and spinal cord DMG and begin to elucidate the heterogeneity observed in antitumour responses.

## Discussion

H3K27M+ DIPG and spinal cord DMGs are extremely aggressive, universally fatal tumours with few therapeutic options. The average life expectancy is 10 months from diagnosis, and the 5-year survival is less than 1%[17]. Palliative radiotherapy is the only established treatment, and neither cytotoxic nor targeted pharmacological approaches have demonstrated improved prognosis to date[18]. Implementation of immunotherapy as treatment for tumours in these precarious neuroanatomical locations is both promising and dangerous. This phase I study was carefully designed to manage potential consequences of inflammation-induced swelling of an already expanded brainstem. The toxicities associated with GD2-CAR T cell infusions were manageable and reversible with intensive supportive care in the inpatient setting. CRS and ICANS were similar to that described with other CAR T cell therapies[19,20], but patients also developed signs and symptoms consistent with CAR T cell-mediated inflammation in sites of CNS disease, which we have termed TIAN. TIAN most often manifested as transient worsening of existing deficits but also resulted in episodes of increased ICP secondary to brainstem oedema and consequent obstructive hydrocephalus, which would have been life-threatening unless immediately and appropriately managed. Our toxicity management algorithm incorporated several pre-emptive measures that enabled safe delivery of this potent therapy. As increasingly efficacious CAR T cell and other immunotherapies are deployed for CNS tumours, TIAN is likely to emerge as an important axis of toxicity in neuro-immuno-oncology.

Two categories of TIAN were evident in these patients: the first category of TIAN relates to ICP and intracranial space constraints secondary to inflammation-induced tissue oedema and/or obstruction of CSF flow. This can require urgent and/or emergent management utilizing principles of neurocritical care. The second category of TIAN relates to primary dysfunction of brain or spinal cord structures affected by inflammation, is typically transient, frequently manifests as worsening of pre-existing symptoms and can often be managed conservatively unless the neurological dysfunction involves critical functions such as respiratory drive.

Although patients developed symptoms of on-tumour neurotoxicity, they did not manifest any signs or symptoms of on-target, off-tumour toxicity. GD2 is expressed on normal neural tissues, including the brain and peripheral nerves, and treatment with the US Food and Drug Administration-approved anti-GD2 antibody dinutuximab for neuroblastoma is associated with transient painful neuropathy in most children[21,22]. None of these four patients developed painful neuropathies or any other clinical or radiographic indication of on-target, off-tumour toxicity. This study adds to the growing evidence that GD2 can be safely targeted with CAR T cells[23–26], and is consistent with evidence that CAR T cells require high antigen density for full effector function[10,13,27,28]. Indeed, autopsy tissue from patient 1 with DIPG confirmed robustly differential GD2 expression levels and T cell infiltration between the tumour and the uninvolved brain.

Although the clinical experience reported here is early and the number of participants treated to date is small, three of four patients derived radiographic and clinical benefit after i.v. administration of GD2-CAR T cells. The improvement observed in neurological function highlights the extent to which this diffusely infiltrating tumour chiefly integrates with and disrupts—rather than destroys—neural circuits[29], underscoring the potential that a tumour cell-specific therapy offers for functional recovery. i.c.v. administration of a second dose of GD2-CAR T cells provided additional radiographic and/or clinical benefit in three of three patients treated. In comparison to i.v. infusions, i.c.v. administrations were associated with less systemic toxicity such as CRS, and correlated with enhanced levels of pro-inflammatory cytokines and reduced immunosuppressive cell populations in CSF. The promising early experience with GD2-CAR T cells for DIPG and spinal cord DMG described here sets the stage for further optimization of this approach for this historically lethal CNS cancer. This clinical trial will continue to treat patients with H3K27M+ DIPG and spinal DMG using GD2-CAR T cell therapy to determine optimal dose, route and schedule, and to determine efficacy.

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

## Methods

### Clinical trial design

This phase I dose-escalation trial of GD2-CAR T cells in children and young adults with pontine and spinal H3K27M⁺ DMG uses a 3 + 3 design with the primary objectives of assessing feasibility of manufacturing, safety and tolerability, and identifying the maximally tolerated dose or recommended phase II dose. Assessment of clinical activity is the secondary objective and identifying correlative biomarkers of response is an exploratory objective. Patients were eligible for enrolment if they had a pathologically confirmed diagnosis of H3K27M-mutated DIPG or spinal cord DMG, had completed standard radiotherapy and were not receiving corticosteroid therapy (additional eligibility and exclusion criteria below).

### Toxicity management

In preclinical studies, a fraction of the mice exhibited brainstem oedema as a result of on-tumour inflammation that led to obstructive hydrocephalus due to the neuroanatomical location of these tumours[2]. We anticipated the development of neurological symptoms related to CAR T cell-mediated inflammation in sites of CNS disease, which we have termed TIAN. To mitigate the risks associated with TIAN, we excluded patients with bulky thalamic or cerebellar tumours, required placement of an Ommaya reservoir in patients with DIPG, conducted both routine and symptom-prompted ICP measurements and instituted a toxicity management plan that included TIAN management (CSF removal via Ommaya, hypertonic saline and/or corticosteroids) and CRS management using anti-cytokine agents (tocilizumab, an IL-6R antagonist, and anakinra, an IL-1R antagonist) and corticosteroids. In addition, the CAR construct incorporated an iCasp9-inducible suicide switch, which could be activated with rimiducid in the event of life-threatening toxicity that is not manageable with the approaches and agents discussed above[30].

### Oversight and informed consent

The clinical study and all amendments were approved by the Stanford University Institutional Review Board. An external data and safety monitoring board (DSMB) reviewed the protocol, amendments and all important patient events and outcomes. Before enrolment in the clinical trial, patients and/or their parents provided written, informed consent, and minor assent was obtained when appropriate. In addition, patients and/or their parents provided written, informed consent for use of photographs and videos that may be used in scientific presentations and publications in print and electronic formats.

### CAR T cell manufacturing

Autologous peripheral blood mononuclear cells (PBMCs) were collected via apheresis and GD2-CAR T cells were manufactured using the closed-system CliniMACs Prodigy (Miltenyi). In brief, CD4 and CD8 T cells were selected and transduced with one bicistronic retroviral vector encoding an iCasp9 domain (Bellicum Pharmaceuticals, Inc.), and a GD2–4-1BB–CD3z CAR (Fig. 1a) containing the 14g2a scFv under control of an MSCV promoter. GD2-CAR T cells were expanded for 7 days in the presence of recombinant human IL-7 and IL-15, plus the addition of the tyrosine kinase inhibitor dasatinib on days 3 and 5 to improve T cell fitness[31].

### CAR T cell administration

Lymphodepleting chemotherapy (cyclophosphamide 500 mg m² daily and fludarabine 25 mg m² daily on days −4, −3 and −2) was administered followed by i.v. CAR T cell infusion on day 0 (Fig. 1b). Patients were monitored closely, including routine ICP measurements, in the inpatient setting through toxicity resolution. For patients receiving second infusions, CAR T cells were infused i.c.v. via the Ommaya. The first patient to be treated with i.c.v. administered CAR T cells (patient 1 with spinal DMG) received a flat dose of 50 million CAR T cells after increased lymphodepleting chemotherapy (cyclophosphamide 600 mg m² daily and fludarabine 30 mg m² daily on days −5, −4, −3 and −2) under a single-patient eIND (see below); subsequent patients received 30 million CAR T cells or the equivalent of their weight-based dose at DL1, whichever was lower, i.c.v. without lymphodepletion.

### Clinical trial eligibility and exclusion criteria

#### Inclusion criteria

(1) Disease status
- (i) Dose-escalation phase and DIPG expansion cohort: tissue diagnosis of H3K27M-mutated DIPG with radiographically evident tumour restricted to the brainstem, or
- (ii) Dose-escalation phase and spinal DMG expansion cohort: tissue diagnosis of H3K27M-mutated DMG of the spinal cord.

(2) Age: greater than or equal to 2 years of age and less than or equal to 30 years of age.

(3) Previous therapy:
- (i) At least 4 weeks following completion of front-line radiation therapy.
- (ii) At least 3 weeks post-chemotherapy or five half-lives, whichever is shorter, must have elapsed since any previous systemic therapy, except for systemic inhibitory/stimulatory immune checkpoint therapy, which requires 3 months.

(4) Performance status: participants over 16 years of age, Karnofsky ≥ 60% or Eastern Cooperative Oncology Group (ECOG) performance status of 0 or 1; participants 16 years of age or younger, Lansky scale ≥ 60%. Participants who are unable to walk because of paralysis, but who are up in a wheelchair, will be considered ambulatory for the purpose of assessing the performance score.

(5) Normal organ and marrow function (supportive care is allowed per institutional standards, that is, filgrastim, transfusion):
- (i) Absolute neutrophil count (ANC) ≥ 1,000 per μl
- (ii) Platelet count ≥ 100,000 per μl
- (iii) Absolute lymphocyte count ≥ 150 per μl
- (iv) Haemoglobin ≥ 8 g dl−1
- (v) Adequate renal, hepatic, pulmonary and cardiac function defined as:
- (vi) Creatinine within institutional norms for age that is, ≤2 mg dl−1 in adults or according to the table below in children younger than 18 years of age) or creatinine clearance (as estimated by Cockcroft Gault equation) ≥60 ml min−1
- (vii) Serum alanine transaminase/aspartate transaminase (ALT/AST) ≤ 3.0 upper limit of normal (ULN; grade 1)
- (viii) Total bilirubin ≤ 1.5 mg dl−1, except in participants with Gilbert's syndrome
- (ix) Cardiac ejection fraction ≥ 45%, no evidence of physiologically significant pericardial effusion as determined by an echocardiogram, and no clinically significant electrocardiogram findings
- (x) Baseline oxygen saturation > 92% on room air

(6) Female individuals of childbearing potential must have a negative serum or urine pregnancy test (female individuals who have undergone surgical sterilization are not considered to be of childbearing potential).

(7) Contraception: participants of childbearing or child-fathering potential must be willing to practice birth control from the time of enrolment on this study and for 4 months after receiving the preparative lymphodepletion regimen or for as long as GD2-CAR T cells are detectable in peripheral blood or CSF.

(8) Ability to give informed consent. Paediatric participants will be included in age appropriate discussion and written assent will be obtained for those 7 years of age or older, when appropriate.

#### Exclusion criteria

(1) Bulky tumour involvement of the cerebellar vermis or hemispheres (pontocerebellar peduncle involvement is allowed), thalamic lesions

that in the investigator's assessment place the participant at unacceptable risk for herniation.

(2) Clinically significant swallowing dysfunction as judged by clinical assessment.

(3) Current systemic corticosteroid therapy.

(4) Previous CAR therapy.

(5) Uncontrolled fungal, bacterial, viral or other infection. Previously diagnosed infection for which the patient continues to receive antimicrobial therapy is permitted if responding to treatment and clinically stable.

(6) Ongoing infection with HIV or hepatitis B virus (HBsAg positive) or hepatitis C virus (anti-HCV positive). A history of hepatitis B or hepatitis C is permitted if the viral load is undetectable per qPCR and/or nucleic acid testing.

(7) Clinically significant systemic illness or medical condition (for example, significant cardiac, pulmonary, hepatic or other organ dysfunction), that in the judgement of the principal investigator is likely to interfere with assessment of safety or efficacy of the investigational regimen and its requirements.

(8) In the investigator's judgement, the participant is unlikely to complete all protocol-required study visits or procedures, including follow-up visits, or comply with the study requirements for participation.

(9) Known sensitivity or allergy to any agents and/or reagents used in this study.

(10) Primary immunodeficiency or history of autoimmune disease (for example, Crohns, rheumatoid arthritis and systemic lupus) requiring systemic immunosuppression and/or systemic disease-modifying agents within the past 2 years.

### Response assessment

Radiographic responses were evaluated by gadolinium-enhanced MRI scans of the brain and/or spinal cord. Because DMGs are diffusely infiltrative of CNS structures and therefore are difficult to measure, volumetric tumour segmentation of T2/FLAIR signal abnormality was performed by a neuroradiologist (K.W.Y.) to measure radiographic change.

Clinical response was assessed by neurological examination. To quantify clinical change, a clinical improvement score (CIS) was calculated. The CIS represents an unweighted quantification of the neurological examination and is conducted by a neurologist who examines the participant before infusion and then at designated timepoints after therapy. For each neurological symptom or sign, one point is assigned. Improvement in a given symptom or sign from pre-infusion baseline add one point to the score, no change adds zero and one point is subtracted for symptom or sign worsening. The reported score is a sum of the positive and negative points. For example, a participant with improved right upper extremity strength and improved left sixth nerve palsy would receive a score of +2. A participant with improved right upper and lower extremity strength, improved left sixth nerve palsy but worsened left facial sensation would also receive a score of +2 (+3 − 1 = +2). The assessment cannot be conducted within 7 days of corticosteroid administration. Sample forms are available in Supplementary Tables 6, 7.

### Cell-free DNA rare mutation detection

Patient CSF samples were collected at pre-determined and trigger timepoints throughout the treatment course. To gently separate the cellular compartment, samples were spun within 1 h of collection at 1,800$g$ for 10 min at 4 °C followed by 20,000$g$ for 10 min. Supernatant was frozen at −80 °C. DNA extraction was performed on 0.5–3 ml CSF supernatant aliquots using the QIAmp Circulating Nucleic Acids Kit (Qiagen) into 25 µl extraction buffer. All samples were assessed for cell-free DNA concentration by nanodrop and fragment size distribution with the BioAnalyzer (Agilent).

Forward and reverse primers and wild-type or mutant probes were designed to optimally detect the H3F3A K27M mutation through digital droplet polymerase chain reaction (ddPCR), and a dilution series of mutant H3K27M g-block against a wild-type H3 background was performed to validate linearity and confirm limit of detection. Seven PCR cycles for pre-amplification were performed on 10 µl of extracted cell-free DNA with Q5 Hot Start High-Fidelity Master Mix (New England Biolabs) for each timepoint. Preamplified reactions were immediately placed on ice after the final extension and diluted with 100 µl TE buffer, pH 8.0, to inactivate the Q5 polymerase. Samples were used immediately for ddPCR (40 cycles; Bio-Rad) and run as 4–6 technical replicates. The raw spectral data were analysed using the Bio-Rad software to plot the signal for mutant and wild-type alleles. All runs contained positive and negative controls. Results were analysed in R to plot variant allele frequency (VAF) as well as mutation count per volume over time.

### qPCR measurement of in vivo GD2-CAR expansion

Patient blood and tumour tissue samples were processed and mononuclear cells were viably cryopreserved. DNA was extracted from whole blood ($2 \times 10^6$–$5 \times 10^6$ PBMCs) using QIAmp DNA Blood Mini Kit (51306, Qiagen) at baseline and multiple timepoints following CAR administration and from tumour tissue samples using DNeasy 96 Blood & Tissue Kit (69582, Qiagen). CAR presence was measured by qPCR using the primer and probe sequences provided as below and in Supplementary Table 8. For the standard curve, a custom Minigene plasmid (IDT) was designed containing a partial GD2–4-1BB–CD3z sequence and a partial albumin sequence, which served as a control for normalization (Supplementary Table 8). The standard curve contained a tenfold serial dilution of plasmid between $5 \times 10^8$ and $5 \times 10^0$ copies. Both plasmid and patient DNA from each timepoint were run in triplicate, with each reaction containing 5 µl of DNA (50 ng total), 200 nM forward and reverse albumin primers (or 300 nM forward and reverse GD2–4-1BB–CD3z primers), 150 nM probe suspended in 10 µl of TaqMan Fast Universal PCR Master Mix (2X), no AmpErase UNG (Thermo Fisher Scientific) and 24.5 µl (albumin) or 22.5 µl (GD2–4-1BB–CD3z) of TE buffer (AM9935, Invitrogen). The Thermo Fisher Scientific QuantStudio 6 Pro Real-time qPCR Instrument was used for qPCR with 20 µl per reaction. The quality metrics for all qPCR standard curve results were $R^2 > 0.95$ and efficiency was 70–110%.

Albumin results from plate normalized to average albumin, then *GD2-CAR* copy number (copies per 50 ng DNA) adjusted to albumin and modified to copies per 100 ng DNA by the following equation: copy number (copies per 100 ng DNA) = 2 × (*GD2-CAR* copy number × (albumin copy number/average albumin)). qPCR reagents included albumin probe (5′-/56-FAM/CCTGTCATG/ZEN/CCCACACAAATCTCTCC/3IABkFQ/-3′), albumin forward primer (5′-GCTGTCATCTCTTGTGGGCTGT-3′), albumin reverse primer (5′-ACTCATGGGAGCTGCTGGTTC-3′), GD2-CAR FAM probe (5′-/56-FAM/TCATGTTGT/ZEN/AGCCGGTGAAGGAGC/3IABkFQ/-3′), GD2-CAR forward primer (5′-CTCTGTGATGATCTCCTGCAA-3′), GD2-CAR reverse primer (5′-CGATCCATTCCAGGCTCTT-3′), and GD2 Albumin Minigene Plasmid (does not include proprietary backbone sequence; 5′-GCTGGCCTTTTGCTCACAAGCTTGGGGTTGCTGTCATCTCTTGTGGGCTGTAATCATCGTCTAGGCTTAAGAGTAATATTGCAAAACCTGTCATGCCCACACAAATCTCTCCCTGGCATTGTTGTCTTTGCAGATGTCAGTGAAAGAGAACCAGCAGCTCCCATG AGTCCCAAGCTATGTTCTTTCCTGCGTTTCTCTGGTGGAACCTGGCGCCTCTGTG ATGATCTCCTGCAAGGCCAGCGGCA GCTCCTTCACCGGCTACAACATGAACTGGG TGCGCCAGAACATCGGCAAGAGCC TGGAATGGATCGGCGCCATCGACCCCTACTACGGCGGCACCAGCT).

### Real-time flow cytometry assay

A high-dimensional (Hi-D) immuno-phenotyping flow cytometry panel was designed for immune profiling of CAR T cells in real time. PBMCs were isolated from fresh whole blood by gradient centrifugation on ficoll (Ficoll-Paque Plus, GE Healthcare, Sigma-Aldrich). Two to five million PBMCs were stained with fixable live/dead aqua (Invitrogen)

amine-reactive viability stain. Cells were then preincubated with Fc block (trustain, BioLegend) for 5 min, then stained at room temperature with the following fluorochrome-conjugated mAb in a 15-colour, 17-parameter staining combination (Supplementary Table 9).

CAR-transduced T cells were used as positive control included in daily staining experiments. Stained and fixed cells were acquired on a LSR (BD BioSciences) five-laser (blue 488 nm, violet 405 nm, UV laser 355 nm, red 640 nm and green 532 nm lasers) analyser. At least $10^6$ cells were acquired unless restricted by the number of cells isolated from 8 ml of whole blood or when acquiring isolated cells from CSF. The assay limit of detection for cells calculated as 1 in $10^4$ of total acquired PBMCs. Representative gating is shown in Extended Data Fig. 10a.

## Luminex cytokines

Patient blood and CSF samples were collected at pre-determined and trigger timepoints throughout treatment. Samples were spun at 250g for 6 min. Supernatant was frozen at −80 °C until batched for assessment. Cytokine assessment was performed by the Immunoassay Team-Human Immune Monitoring Center at Stanford University. Panels include Luminex-EMD Millipore HIMC H80 (panel 1 is Milliplex HCYTA-60K-PX48; panel 2 is Milliplex HCP2MAG-62K-PX23; panel 3 includes the Milliplex HSP1MAG-63K-06 and HADCYMAG-61K-03 (resistin, leptin and HGF) to generate a 9 plex) and TGFβ (TGFBMAG-64K-03). Kits were purchased from EMD Millipore Corporation and used according to the manufacturer's recommendations with modifications described. The assay setup followed recommended protocol. In brief, samples were diluted threefold (panels 1 and 2) or tenfold (panel 3). Of the diluted sample, 25 µl was mixed with antibody-linked magnetic beads in a 96-well plate and incubated overnight at 4 °C with shaking. Cold and room temperature incubation steps were performed on an orbital shaker at 500–600 rpm. Plates were washed twice with wash buffer in a Biotek ELx405 washer (BioTek Instruments). Following 1-h incubation at room temperature with biotinylated detection antibody, streptavidin–phycoerythrin (PE) was added for 30 min with shaking. Plates were washed as described above and PBS was added to wells for reading in the Luminex FlexMap3D Instrument with a lower bound of 50 beads per sample per cytokine. Custom Assay Chex control beads were purchased and added to all wells (Radix Biosolutions). All wells collected met quality control metrics with a bead count >50. Data are represented in pg ml$^{-1}$ based on standard curves or heat maps of fold change from baseline timepoint. All samples were run in technical duplicate.

## Single-cell RNA sequencing

Single-cell RNA sequencing (scRNA-seq) was performed using 5′ v2 Single Cell Immune Profiling technology (10X Genomics) according to the manufacturer's protocol. In brief, cells from CSF samples collected at indicated timepoints before and after CAR T cell administration, as well as sorted CAR$^+$ and CAR$^-$ cells from the CAR T cell infusion products, were counted, resuspended to 700–1,200 cells per µl, and captured using Single Cell Chip A on the 10x Chromium Controller (10X Genomics) to generate gel bead-in emulsions (GEMs). Reverse transcription inside GEMs was performed using a C1000 Touch Thermal Cycler (Bio-Rad). Barcoded complementary DNA (cDNA) was recovered through post-GEM-RT cleanup and PCR amplification. Recovered cDNA was amplified and used to construct 5′ whole-transcriptome libraries. Quality of cDNA and each library was assessed using Agilent 2100 Bioanalyzer. The libraries were indexed using a Chromium i7 Sample Index Kit, pooled and sequenced on NovaSeq 6000 System (Illumina) by Novogene.

Raw sequencing data were processed using the Cell Ranger software version 6.0.0 or higher (10X Genomics). Sequencer's base call files (BCLs) were demultiplexed into FASTQ files using the cellranger mkfastq pipeline. To optimize CAR sequence detection in the scRNA-seq data, we probed the FASTQ files using SeqKit[32] in 50-base-pair segments from the GD2-CAR plasmid sequence. Once we identified the segments with the most hits, we created a custom reference by appending these segments to the GRCh38 5.0.0 human genome reference with the cellranger mkref pipeline. FASTQ files were aligned to this custom reference using the cellranger count pipeline.

Unique molecular identifier (UMI) count matrices from Cell Ranger were analysed using Seurat[33]. Dead cells and cell debris with more than 15% of UMI counts mapping to mitochondrial genes or less than 300 genes detected were excluded from the analysis. Cell doublets containing more than 10,000 genes or more than 50,000 UMI counts were also excluded. Next, scRNA-seq data were subsampled to 5,000 cells per sample, and all data were normalized together to the sequencing depth using the SCTransform pipeline[34]. Data between patients were integrated using reciprocal principal component analysis (RPCA). PCA was performed using all variable genes with the exception of TCR and BCR genes to prevent clonotypes from driving the final layout. In addition, a set of curated genes relevant to T cell function or cell-type identification were included into the list of variable genes. UMAP embedding was performed using the first 50 principal components. Cell types were assigned to all cells based on canonical lineage marker expression. Cell-type doublets ('lineage doublets') were identified as cells expressing markers of more than one lineage: T cells (CD2, CD3E, CD3D, CD3G, CD247, CD7 and GD2-CAR), B cells (CD19, CD22 and PAX5), microglia/myeloid cells (CD14, CD68, CD163, CSF1R and AIF1) or astrocytes (GFAP). Differential expression analysis was performed on SCT counts modelled as a negative binomial distribution using the FindMarkers pipeline.

Further analysis of patient 1 with DIPG and other participant CSF samples was performed using the R-based 'Seurat' package[35] to assess transcriptional differences preceding and during CAR T cell administration. First, the large Seurat object was subset into various smaller objects to enable multiple comparisons across different DIPG CSF samples (see Fig. 4). Seurat objects subsequently underwent scaling and dimensionality reduction based on the UMAP algorithm[36]. Myeloid fractions of the dataset were identified using the 'AddModuleScore' function in Seurat to isolate cells with relatively high expression of myeloid genes (AIF1, CSF1R, CX3CR1, CD14, CD68 and CD163), compared to their expression of B cell-related (CD19, CD22 and PAX5) and T cell-related (CD2, CD3E, CD3D, CD3G, CD247 and CD7) genes. Scoring of cells based on disease-associated myeloid (DAM) or myeloid-derived suppressor cell (MDSC) gene expression was implemented using the 'AddModuleScore' and 'FeaturePlot' functions in Seurat. DAM genes included SPP1, GPNMB, IGF1, CLEC7A, LPL, CD9, CD63, LGALS3, FABP5, ITGAX, APOE and TYROBP, based on previous noteworthy characterizations of myeloid cells in disease[15,16,37,38]. MDSC genes included CD33, CD14, CD15 and IL4RA[39]. Volcano plot analysis was conducted using the R-based 'EnhancedVolcano' package[40]. Gene ontology (GO) analysis was performed using the R-based 'ReactomePA' package[41].

Single-cell expression scores were computed in a similar way as previously described[42]. Given a set of genes ($Gj$) for a gene set (for example, a DAM signature), a score, $SCj(i)$, which quantifies the scaled expression (Z-score) of $Gj$ for each cell $i$, was computed as the average scaled expression (Er) of the genes in $Gj$, compared to the average scaled expression of a control gene set Gcont: $SC(i) = \text{average}[\text{Er}(G, i)] - \text{average}[\text{Er}(Gcont, i)]$. For each gene of the gene set, its control gene set contains 100 genes with the most similar aggregated expression level to that gene. Therefore, the control gene set represents a 100-folder larger but comparable distribution of expression levels to that of the considered gene set.

Graph-based clustering with data integration was adapted to identify cellular clusters. We selected highly variable genes (HVGs) using the FindVariableFeatures function in Seurat and used the scaled expression values of these HVGs for PCA. To disentangle sample-specific biological variations from cell subpopulation-specific variations and integrate multiple samples, we applied a linear adjustment method called Harmony to the first 30 principle components with default parameters to

generate a corrected embedding[43]. We chose the first 20 Harmony corrected dimensions for constructing UMAP embeddings by RunUMAP in Seurat and clustered cells by the Louvain algorithm based FindClusters in Seurat. Cells that were from different samples, but expressed similar gene programmes, were well mixed. We next identified differentially expressed genes (DEGs) by FindAllMarkers in Seurat and tested genes that were detected in a minimum of 30% of the cells within each cluster and that showed at least a 0.5-fold mean log difference. We utilized Wilcoxon rank-sum test with Bonferroni correction for multiple testing and only kept genes with adjusted $P < 0.05$. Cell clusters were annotated with manual inspection of their top DEGs. In addition, we tested for enrichment of described gene sets (GO biological processes) and compared expression programmes of each cell cluster with those of a published glioblastoma-associated myeloid cell dataset[44].

## Histology, immunohistochemistry, immunofluorescence and RNAscope

Immunohistochemistry (other than for GD2 and H3K27M) was performed on formalin-fixed paraffin-embedded tissue sections per standard protocols including deparaffinization, antigen retrieval, incubation with primary antibody, and detection per the manufacturers' instructions. The following antibodies were used: CD3 (790–4341, Ventana; rabbit polyclonal, prediluted), CD4 (NCL-CD4-368, Leica (Novocastra); mouse monoclonal, 1:40 dilution), CD8 (M7103, Dako; monoclonal mouse, 1:400 dilution) and CD163 (760–4437, Ventana; monoclonal mouse, prediluted). Stains for CD3 and CD163 were performed on a Ventana BenchMark Ultra automated stainer using CC1 antigen retrieval. Stains for CD4 and CD8 were performed on a Leica Bond automated stainer using ER2 antigen retrieval.

For H3K27M and GD2 immunohistochemistry, primary tumour samples from patients were transferred to cryomolds and embedded in optimal-cutting temperature (OCT) compound (TissueTek). Cryosections (10 µm) were generated on a cryostat (Leica). Tissue was fixed with 4% PFA at 4 °C for 20 min then washed and endogenous peroxidase activity was neutralized (Bloxall, Vector Laboratories; 10 min at room temperature) before permeabilization (0.3% Triton X-100, TBS, 15 min at room temperature) and blocking (5% horse serum, Vector Laboratories; 20 min at room temperature). Sequential double-staining immunohistochemistry was conducted for H3K27M (ab190631, Abcam; 1:1,000, 1 h at room temperature) and GD2 (14g2a, BD; 1:500, 1 h at room temperature). H3K27M was developed with a peroxidase secondary (ImmPRESS VR anti-rabbit IgG, Vector Laboratories; 30 min at room temperature) and DAB substrate (BD). After quenching the DAB substrate development in TBS and staining with the 14g2a primary antibody, the GD2 signal was developed using a polymer-based alkaline phosphatase secondary antibody (ImmPRESS AP anti-mouse IgG, Vector Laboratories; 30 min at room temperature) and blue alkaline phosphatase substrate (Vector Blue AP substrate kit, Vector Laboratories; 5 min at room temperature). Alkaline phosphatase development was quenched in TBS, and samples were mounted and imaged by light microscopy (Zeiss Axio Imager M2).

For immunofluorescence, primary tumour samples from patients were fixed in 4% PFA overnight and then transferred to 30% sucrose until samples sunk (3–4 days). Serial 40-µm cryosections were generated on an automatic freezing microtome (HM450, Thermo Fisher) then incubated with 3% normal donkey serum in 0.3% Triton X-100 in TBS blocking solution at 1 h at room temperature. Sections were stained overnight at 4 °C in primary antibody. Antibodies used were mouse anti-H3K27M (ab190631, Abcam; 1:1000) and rabbit anti-IBA1 (019–19741, Wako; 1:500) diluted in 1% normal donkey serum in 0.3% Triton X-100 in TBS blocking solution. Sections were incubated in secondary antibody conjugated with either 594 or 488 for 2 h at room temperature (715-585-150 and 711-545-152, Jackson Immunoresearch; 1:500). Samples were mounted with ProLong Gold mounting medium (P36930, Life Technologies) and were imaged using confocal microscopy (LSM710,

Zeiss). For RNAscope *in situ* hybridization to visualize the *GD2-CAR* construct, primary tumor samples from patients were transferred to cryomolds and embedded in optimal-cutting temperature (OCT) compound (TissueTek). 10-µm cryosections were generated on a cryostat (Leica). Slides were fixed with cold 4% PFA for 15 minutes at 4 °C and then dehydrated in increasing ethanol rinses. To prepare for staining, tissue was pretreated with RNAscope hydrogen peroxide for 10 min at room temperature, then treated with RNAscope protease IV and incubated for 30 min at room temperature (ACD 322381). Tissue was then stained following the kit protocol for RNAscope 2.5 HD Duplex Assay (ACD, 322500). To stain control cells for *GD2-CAR* RNAscope, GD2-CAR T cells were thawed from frozen and prepared for staining according to the RNAscope Multiplex v2 Assay (ACD document MK-50 010). In short, cells were seeded in growth medium on chamber slides for 24 h and then fixed with 4% PFA for 30 min at room temperature. Cells were then dehydrated with ethanal and stored in 100% ethanol at −20 °C until staining. On the day of staining, cells were rehydrated with ethanol rinses and then pretreated with RNAscope hydrogen peroxide for 10 min at room temperature followed by RNAscope protease III diluted 1:15 with PBS for 10 min at room temperature (ACD 322381) and then stained alongside the tissue samples. A custom probe was designed to target the *GD2-CAR* construct using the following sequence: AUGCUGCUGCUCGUGACAUCUCUGCUGCUGUGCGAGCUGCCCCACC CCGCCUUUCUGCUGAUCCCCGAUAUCCUGCUGACCCAGACCCCUCUG AGCCUGCCUGUGUCUCUGGGGCGAUCAGGCCAGCAUCAGCUGCAGAU CCAGCCAGAGCCUGGUGCACCGGAACGGCAACACCUACCUGCACU GGUAUCUGCAGAAGCCCGGCCAGAGCCCCAAGCUGCUGAUUCACAAG GUGUCCAACCGGUUCAGCGGCGUGCCCGACAGAUUUUCGGCAGC GGCUCCGGCACCGACUUCACCCUGAAGAUCAGCCGGGUGGAAGCCGA GGACCUGGGCGUGUACUUCUGCAGCCAGUCCACCCACGUGCCCCCC CUGACAUUUGGCGCCGGAACAAAGCUGGAACUGAAGGGCAGCACA AGCGGCAGCGGCAAGCCUGGAUCUGGCGAGGGAAGCACCAAGGGCGA AGUGAAGCUGCAGCAGAGCGGCCCCUCUCUGGUGGAACCUGGCGCC UCUGUGAUGAUCUCCUGCAAGGCCAGCGGCAGCUCCUUCACCGGCU ACAACAUGAACUGGGUGCGCCAGAACAUCGGCAAGAGCCUGGAAUGG AUCGGCGCCAUCGACCCCUACUACGGCGGCACCAGCUACAACCAGA AGUUCAAGGGCAGAGCCACCCUGACCGUGGACAAGAGCAGCUCCACC GCCUACAUGCACCUGAAGUCCCUGACCAGCGAGGACAGCGCCGUGUA CUACUGCGUGUCCGGCAUGGAAUACUGGGGCCAGGGCACAAGCGUG ACCGUGUCCUCUGCGGCCGCAACCACGACGCCAGCGCCGCGACCACC AACACCGGCGCCCACCAUCGCGUCGCAGCCCCUGUCCCUGCGCCCA GAGGCGUGCCGGCCAGCGGCGGGGGGCGCAGUGCACACGAGGGG GCUGGACUUCGCCUGUGAUAUCUACAUCUGGGCGCCCUUGGCCGGG ACUUGUGGGGUCCUUCUCCUGUCACUGGUUAUCACCCUUUACUGC AAACGGGGCAGAAAGAAACUCCUGUAUAUAUCAAACAACCAUUUAUG AGACCAGUACAAACUACUCAAGAGGAAAUGGCUGUAGCUGCCGAUUU CCAGAAGAAGAAGAAGGAGGAUGUAACUGAGAGUGAAGUUCAGCAGG AGCGCAGACGCCCCCGCGUACCAGCAGGGCCAGAACCAGCUCUAUA ACGAGCUCAAUCUAGGACGAAGAGAGGAGUACGAUGUUUUGGACAAG AGACGUGGCCGGGACCCUAGAUGGGGGGAAAGCCGAGAAGGAAGA ACCCUCAGAAGGCCUGUACAAUGAACUGCAGAAAGAUAAGAUGGCG GAGGCCUACAGUGAGAUUGGGAUGAAAGGCGAGCGCCGGAGGGGC AAGGGGCACGAUGGCCUUUACCAGGGUCUCAGUACAGCCACCAAGG ACACCUACGACGCCCUUCACAUGCAGGCCCUGCCCCCUCGC. After probe hybridization, amplification and detection according to the kit manual, slides were counterstained with 50% hematoxylin for 30 sec, followed by 0.02% ammonia water for blueing. Slides were dried at 60 °C for 15 min, then mounted and imaged by light microscopy (Zeiss Axio Imager M2).

## Flow cytometry of resected tumour tissue

Tumour resection material was digested into single-cell suspensions as previously described[45]. Single-cell suspensions were stained with the following antibodies (Supplementary Table 10) and gating was performed using fluorescence minus one controls (FMO) on a BD LSRFortessa. DIPG

cells were gated to evaluate GD2 expression in the CD45−, B7-H3+ population. The gating strategy is depicted in Extended Data Fig. 10b.

## Reporting summary

Further information on research design is available in the Nature Research Reporting Summary linked to this paper.

## Data availability

All raw data are provided in the source data files. The single-cell RNA sequencing data have been uploaded to the Gene Expression Omnibus (GSE186802). Source data are provided with this paper.

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

**Acknowledgements** We are grateful to our courageous patients and their families; A. Quinones-Hinojosa, K. Jaeckle, H. Tran, D. Poon, M. Casazza, D. Brown, S. Herrera, R. Taiwo, R. Jamiolkowski, A. Wu, S. Vaca, D. Herrick, D. Purger, D. Hong, A. Cluster, D. Limbrick, E. Mayne, C. Tesi-Rocha, C. Wusthoff, S. Lee, F. Baumer, J. Klotz, T. Nguyen and many others on the clinical teams for collaboration in the care of these patients; J. Senesac and Bellicum for providing study-enabling materials in the form of the viral vector and the AP1903 supply; J. S. Tamaresis for statistical support in the trial design; Y. Pan for help with figure formatting; S. Marsh and B. Stevens for input on myeloid cell transcriptional signatures; N. Lepori-Bui, T. Tutt and A.-L. Gramstrup Petersen for their expertise in cell formulation; and the Stanford Human Immune Monitoring Core (HIMC) for performing cytokine assays. This work was supported by the National Institutes of Health R01 CA263500-01 (C.L.M. and M.M.), the Parker Institute for Cancer Immunotherapy (C.L.M. and R.G.M.), CureSearch (C.L.M., M.M. and R.G.M.), a St. Baldrick's Foundation - Stand Up 2 Cancer Pediatric Cancer Dream Team translational research grant (SU2CAACR-DT1113; C.L.M., R.G.M., S.R. and M.M.), the Stella S. Jones Foundation (M.M.), The V Foundation (R.G.M.), NIH Director's Pioneer Award (DP1NS111132; M.M.), NIH Director's New Innovator Award (DP2-CA272092; R.G.M.), ChadTough Defeat DIPG (J. Mahdi), Waxman Family Research Fund (A.C.G.) and the Virginia and D.K. Ludwig Fund for Cancer Research (C.L.M. and M.M.). The St. Baldrick's Foundation partners with Stand Up 2 Cancer, a programme of the Entertainment Industry Foundation administered by the American Association for Cancer Research.

**Author contributions** M.M. is the principal investigator of the trial. C.L.M. is the Investigational New Drug (IND) holder. C.W.M., R.G.M., C.L.M. and M.M. conceived the project. S.M., E.E. and C.B. oversaw regulatory affairs. S.M., E.E., M.M., C.L.M., C.W.M., R.G.M., S. Partap, C.J.C., L.R., T.T.C., K.E.W. and G.G. planned, designed and/or wrote the clinical trial, amendments and treatment protocols. K.E.W. and M.M. designed the clinical improvement scale. S. Patel, H.C. and S.A.F. performed process development for cellular manufacturing and supervised cellular manufacturing. R.G.M., S.R., L.M.S., R.M.R., V.B, J. Moon, A.R., E.H.N., C.B., J. Mahdi, C.E., S.G., L.R., T.T.C., S. Partap, P.G.F., C.J.C., G.G., K.L.D., C.L.M. and M.M. participated in patient care. K.W.Y. read radiographs and performed volumetric analysis of MRI scans. I.J.C., M.C.R., C.B., J. Moon., M.K. and M.F. participated in the collection and/or processing of patient samples. A.C.G., V.B., R.G.M., Z.E., W.R., S.K. and B.S. performed correlative studies. A.C.G. performed GD2 tissue staining and designed and carried out RNAscope in situ hybridization. V.B. designed and carried out the ctDNA assay. S.R., L.J., R.M., Z.G., A.Y.M., S.M.G. and M.G.F. analysed scRNA-seq data. S. Prabhu., S.R., V.B., A.C.G., R.G.M., M.M. and C.L.M. analysed other correlative studies. A.M.S.T. and H.V. performed the brain autopsy for patient 1 with DIPG and performed neuropathological examination and immunohistochemical analyses on the autopsy tissue. R.G.M., S.R., M.M. and C.L.M. prepared figures and wrote the manuscript. M.M. and C.L.M. supervised all aspects of the work.

**Competing interests** Stanford University is in the process of applying for a patent application covering treatment of H3K27M-mutated gliomas with GD2-CAR T cells that lists M.M., C.L.M., R.G.M and C.W.M. as inventors. C.L.M. is a cofounder and holds equity in Lyell Immunopharma and Syncopation Life Sciences, which are developing CAR-based therapies, Red Tree Venture Capital, Ensme and Mammoth and consults for Lyell, Syncopation, Red Tree, NeoImmune Tech, Apricity, Nektar, Immatics, Ensoma and Mammoth. R.G.M. is a cofounder of and holds equity in Syncopation Life Sciences; he is also a consultant for Lyell Immunopharma, Syncopation Life Sciences, NKarta, Gamma Delta Therapeutics, Aptorum Group, Illumina Radiopharmaceuticals, ImmunAI, Arovella Therapeutics and Zai Lab. M.M. is on the scientific advisory board for Cygnal Therapeutics.

**Additional information**
**Correspondence and requests for materials** should be addressed to Crystal L. Mackall or Michelle Monje.

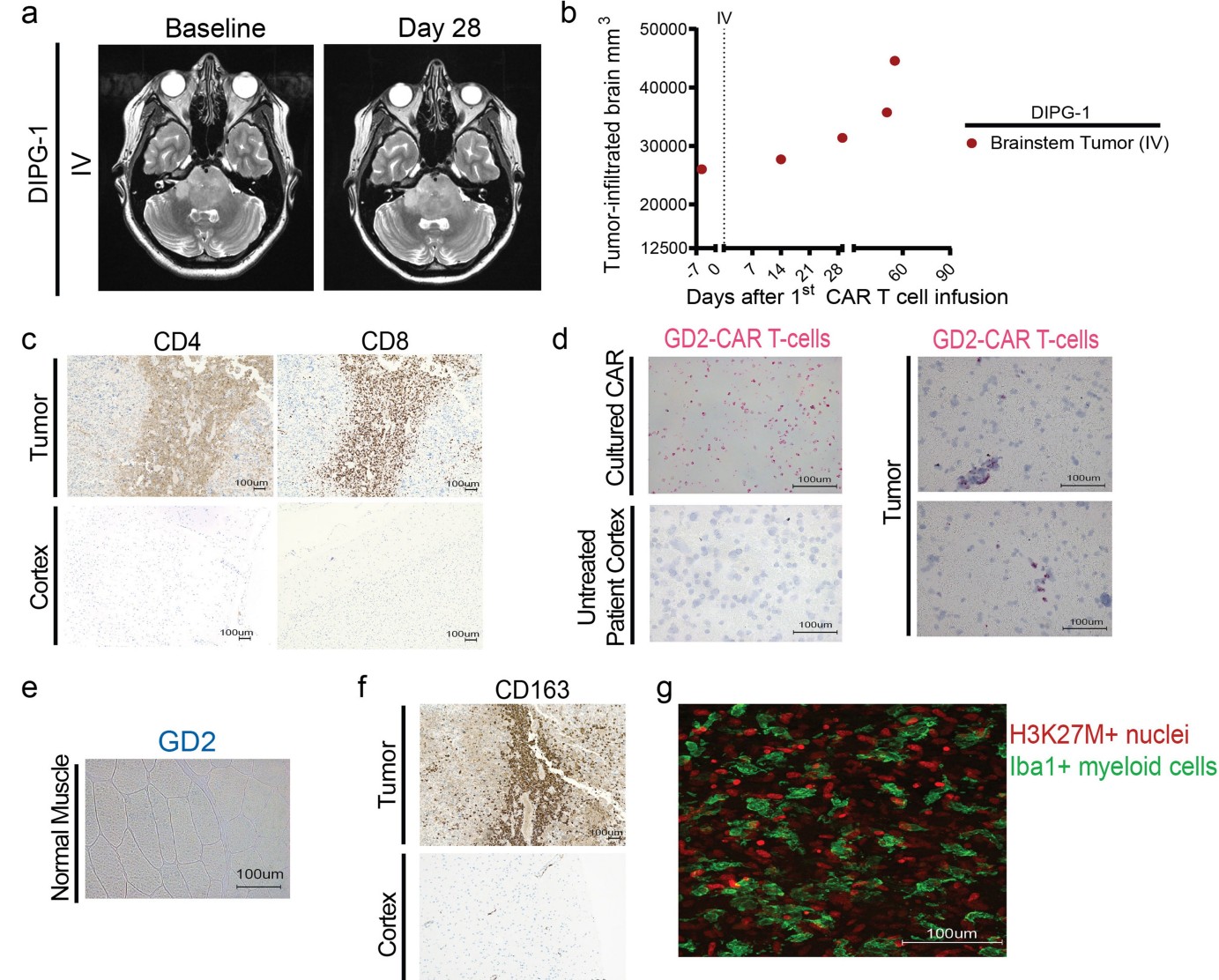

**Extended Data Fig. 1 | Additional correlative findings and imaging for DIPG-1. a**, MRI images (axial T2) of participant 1 with DIPG (DIPG-1) prior to and Day+28 following GD2-CAR T infusion showed no improvement. **b**, Tumour volume change over time in DIPG-1. **c**, Post-mortem pons tumour tissue of DIPG-1 shows evidence of CD4+ and CD8+ T-cells by immunohistochemistry (brown). Unaffected cortex CD4+ staining depicts rare leptomeningeal vascular CD4+ cells and serves as an internal positive control. **d**, RNAscope probe against the *GD2-CAR* construct mRNA identifies GD2-CAR T cells (*GD2-CAR* mRNA puncta, pink). Positive control GD2-CAR T cells in tissue culture and negative control cortex from a DIPG patient not treated with CAR T cells were used to validate the RNAscope probe. *GD2-CAR* mRNA expression detected by RNAscope identified GD2-CAR T cells in tumour tissue from DIPG-1. *GD2-CAR* mRNA puncta = pink, hematoxylin counterstain for all cells = blue. **e**, Normal human muscle tissue immunostained for GD2 as negative control for GD2 antigen immunohistochemistry. **f**, CD163+ myeloid cells (brown) in tumor tissue of DIPG-1 by immunohistochemistry. Unaffected cortex CD163 immunostaining demonstrates microglial cells in their "resting" perivascular location. CD163+ reactive microglia/macrophages were not evident in the normal cortex of DIPG-1. **g**, Confocal microscopy of DIPG-1 tumour tissue obtained at autopsy demonstrates significant myeloid cells (Iba1+ cells, green) infiltrating the tumor (H3K27M+ cells, red). Scale bars = 100 micrometers.

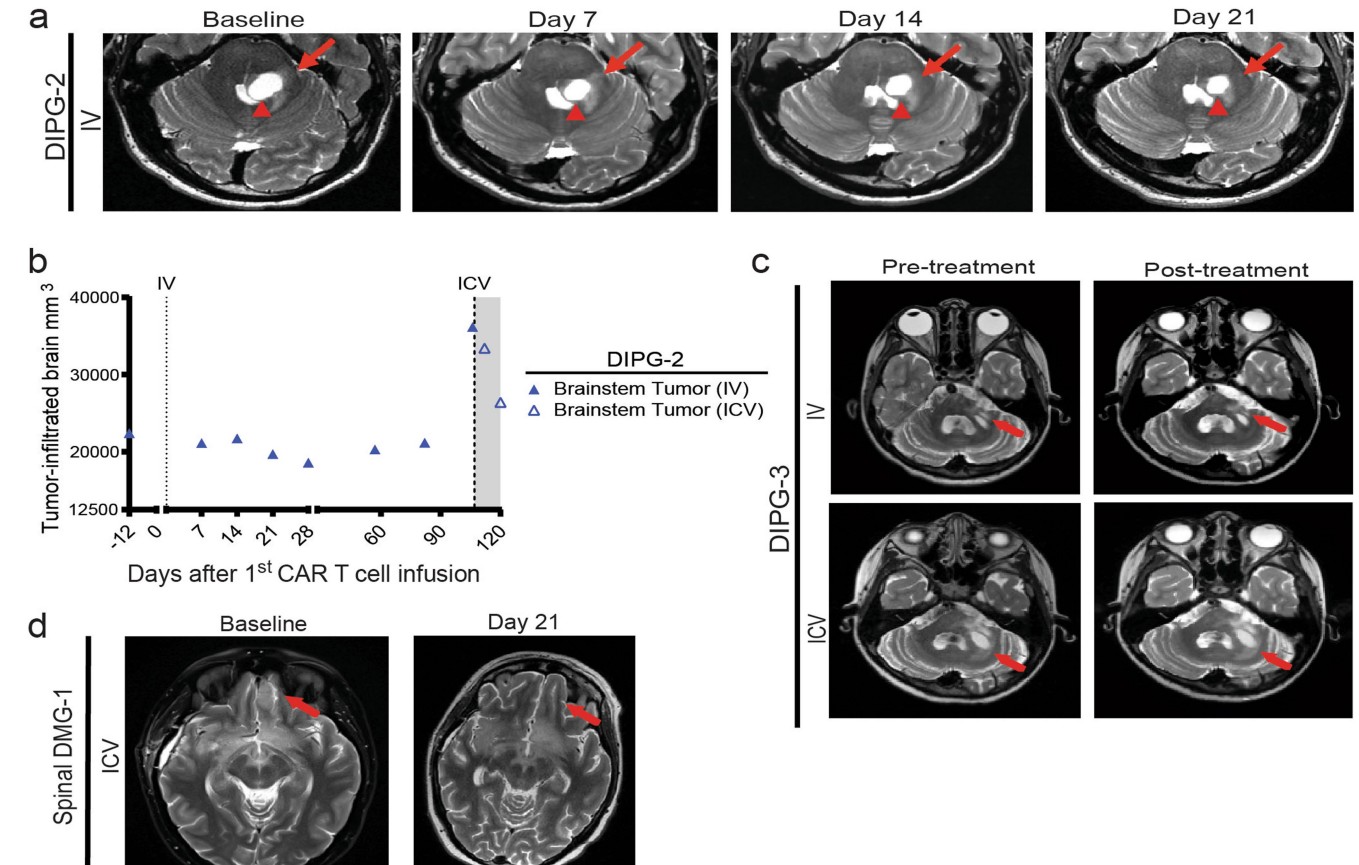

**Extended Data Fig. 2 | Additional MRI findings. a**, Participant 2 with DIPG (DIPG-2) MRI images (axial T2) at the level of the mid-pons demonstrate T2 signal abnormality in the trigeminal nucleus (red arrow) that worsened (increased T2 signal, consistent with pseudoprogression) at Day+7 and progressively decreased on the Day+14 and Day+21 scans. Together with this T2 signal change, trigeminal function (muscles of mastication and left facial sensation) worsened around Day+7 and then progressively improved clinically. The size of the intra-tumoral cyst near the trigeminal nucleus decreased in size after treatment (red arrowhead) and the size and shape of the fourth ventricle normalized. **b**, Tumor volume change over time in DIPG-2. **c**, MRIs (axial T2) of cerebellum from participant 3 with DIPG (DIPG-3) shows increased disease in cerebellum over time, despite GD2-CAR T cell treatments. Disease in the cerebellar peduncle was relatively stable at 1 month post-IV infusion, but by the time of ICV infusion (new ICV baseline, bottom row) the cerebellar peduncle and cerebellar disease had increased, consistent with tumour progression at this site. **d**, Patient 1 with spinal cord DMG (Spinal DMG-1) MRI brain (axial T2) prior to and 21 days after ICV GD2-CAR T cell administration. Note the extensive spread of tumour throughout brain, which reduced following ICV treatment (for example, see arrows at the left inferior frontal lobe highlighting decreased infiltrative disease at Day 21).

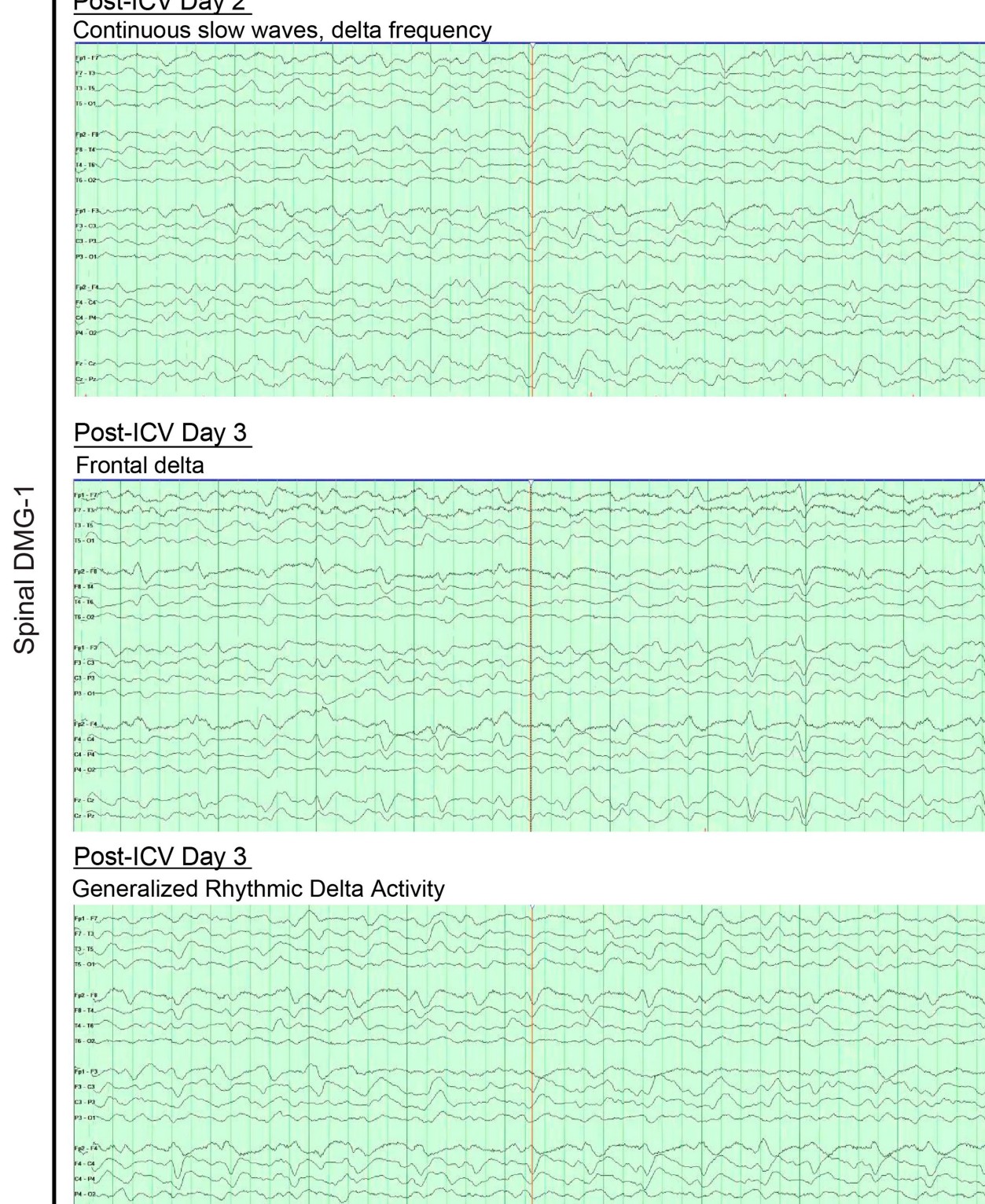

## Post-ICV Day 2
Continuous slow waves, delta frequency

## Post-ICV Day 3
Frontal delta

## Post-ICV Day 3
Generalized Rhythmic Delta Activity

Spinal DMG-1

**Extended Data Fig. 3 | Spinal DMG-1 EEG during episode of encephalopathy.** Electroencephalogram (EEG) demonstrates diffuse slowing with triphasic waves on Day 2-3 following ICV GD2-CAR T cell administration in Spinal DMG-1. Consistent with this EEG pattern that is often observed with reversible toxic/metabolic/inflammatory encephalopathy, mental status returned to baseline over the course of days.

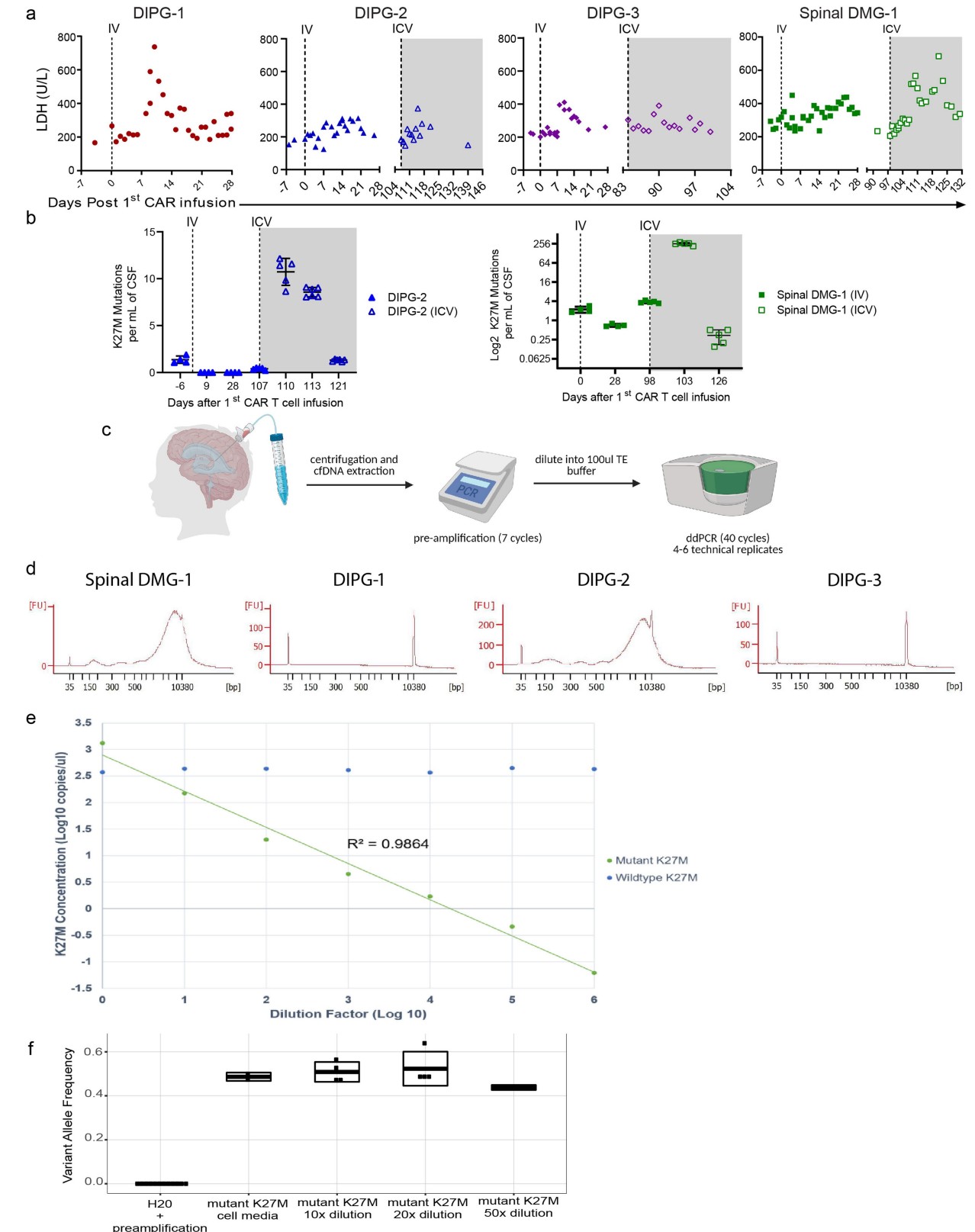

**Extended Data Fig. 4 |** See next page for caption.

**Extended Data Fig. 4 | Peripheral blood LDH kinetics, CSF cell-free tumor DNA (ctDNA) findings, and ctDNA validation. a**, Elevated Lactate Dehydrogenase (LDH) levels approximately 7–14 days following GD2-CAR T cell infusions (either IV or ICV) in N = 4 patients. Filled dot=IV, open dot=ICV. **b**, Cell-free tumor DNA (ctDNA) from patient CSF was evaluated using digital droplet PCR for the H3K27M mutation in the *H3F3A* gene. N=at least 4 technical replicates represented as Log2 of mutations per mL CSF ± SEM. Each point equals one technical replicate at the indicated timepoint; Filled dot=IV, open dot=ICV. DIPG-2 and Spinal DMG-1 demonstrated increasing cfDNA levels directly following GD2-CAR T cell ICV infusion (p < 0.0001 calculated by t-test). All available CSF samples from patients with adequate DNA extraction are represented here. DIPG-1 and DIPG-3 exhibited below the limit of detection levels and are not shown (limit of detection = 1 mutation per mL). **c**, Workflow for cell-free tumor DNA (ctDNA) assay. Cell-free DNA (cfDNA) was extracted from 1–4ml of CSF and subjected to 7 cycles pre-amplification followed by 40 cycles of digital droplet PCR (ddPCR). Schematic created with BioRender.com.

**d**, Representative bioanalyzer traces of patients demonstrate adequate cfDNA quantities in Spinal DMG-1 and DIPG-2 without detectable cfDNA in DIPG-1 and DIPG-3 (i.e. there was not enough total cell-free DNA to run the assay). **e**, Validation of ddPCR assay through serial dilution of mutant H3K27M gBlock (IDT) against a fixed background of wild type H3K27M gBlock demonstrates linear mutant H3K27M detection. Sample concentrations (copies/ul) and dilution factors are plotted at Log10 scale. The QuantaSoft Pro Software calculates the starting concentration of each target DNA molecule by modeling as a Poisson distribution; the formula used for Poisson modeling is: Copies per droplet = -ln(1-p) where p = fraction of positive droplets. **f**, Inclusion of negative and positive controls for each ddPCR assay was performed alongside patient samples. Pre-amplified water served as a negative control for every run. Conditioned medium from H3K27M-mutated DIPG cell cultures was used as a positive control. Serial dilutions of H3K27M-mutated DIPG cell culture medium demonstrate a reproducible variant allele frequency (VAF) of the positive control.

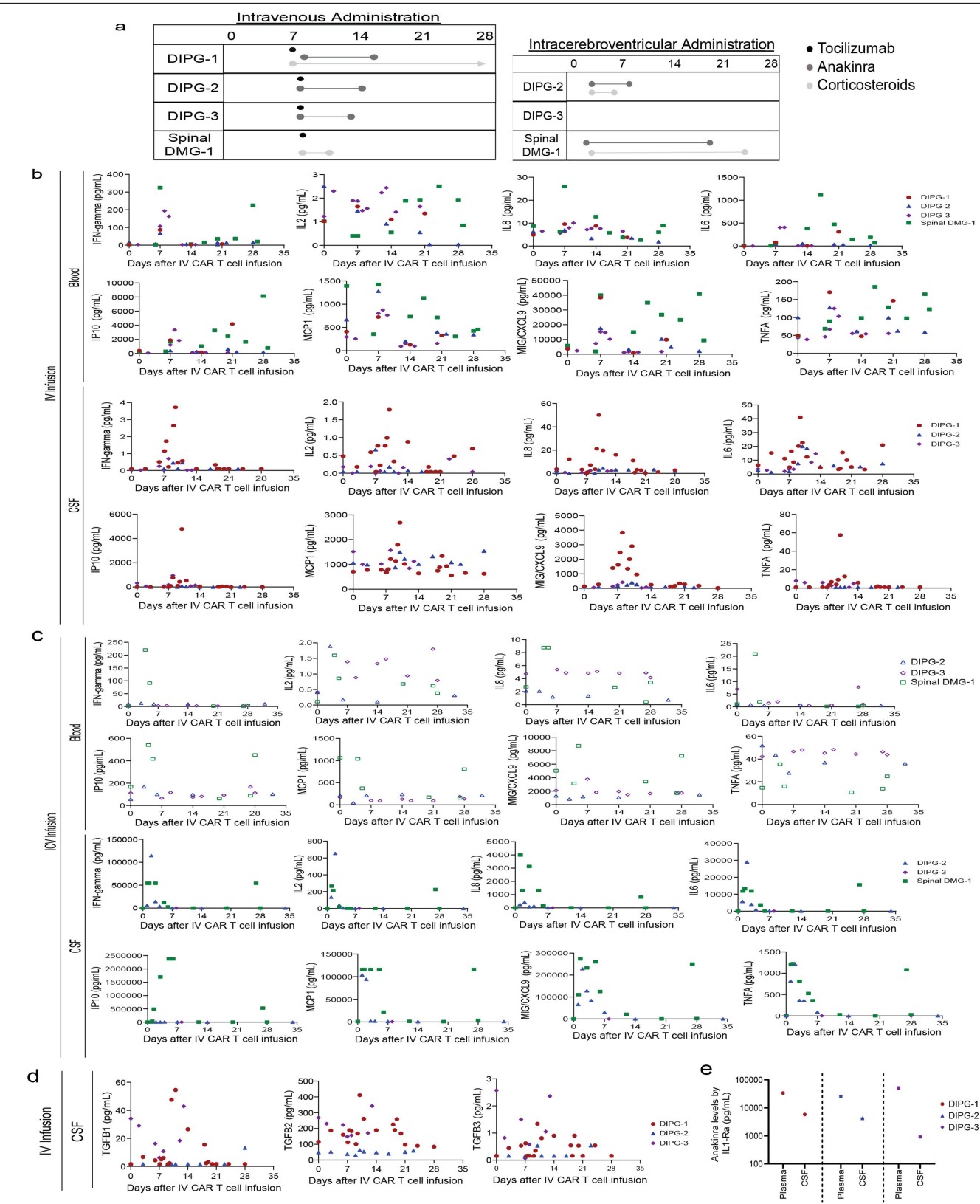

**Extended Data Fig. 5** | See next page for caption.

**Extended Data Fig. 5 | Absolute values of pro-inflammatory and suppressive cytokines in pg/mL and anakinra levels measured by IL1-Ra in CSF and serum. a**, Time-course of anti-inflammatory agents (tocilizumab, anakinra and corticosteroids) administered to patients following CAR T cell infusions represented graphically to provide context for interpreting cytokine levels at various timepoints. **b**–**d**, The same data in absolute values that is represented in heatmap form in Fig. 4. Each data point represents the pg/ml value of the indicated cytokine at the timepoint following GD2-CAR T cell infusion. **b**, Pro-inflammatory cytokine levels in blood and cerebrospinal fluid (CSF) following intravenous (IV) administration. N = 4 patients (blood) and 3 patients (CSF). **c**, Pro-inflammatory cytokine levels in blood and cerebrospinal fluid (CSF) following intra-cerebroventricular (ICV) administration. N = 3 patients. **d**, Immune-suppressive TGFβ cytokine levels in the CSF following IV CAR T cell administration. N = 3 patients. **e**, Peak serum and CSF IL1-Ra levels during anakinra treatment. Anakinra is recombinant IL1-Ra. CSF levels of anakinra, which can cross the blood-brain barrier, were approximately one tenth those of serum levels. Data generated by Luminex multi-plex cytokine analysis of patient blood plasma and CSF samples. N = 3 patients. **b**–**e**, Each data point represents the mean of two technical replicates. Red=DIPG-1, Blue=DIPG-2, Purple= DIPG-3, Green=Spinal DMG-1. Filled dot=IV, open dot=ICV.

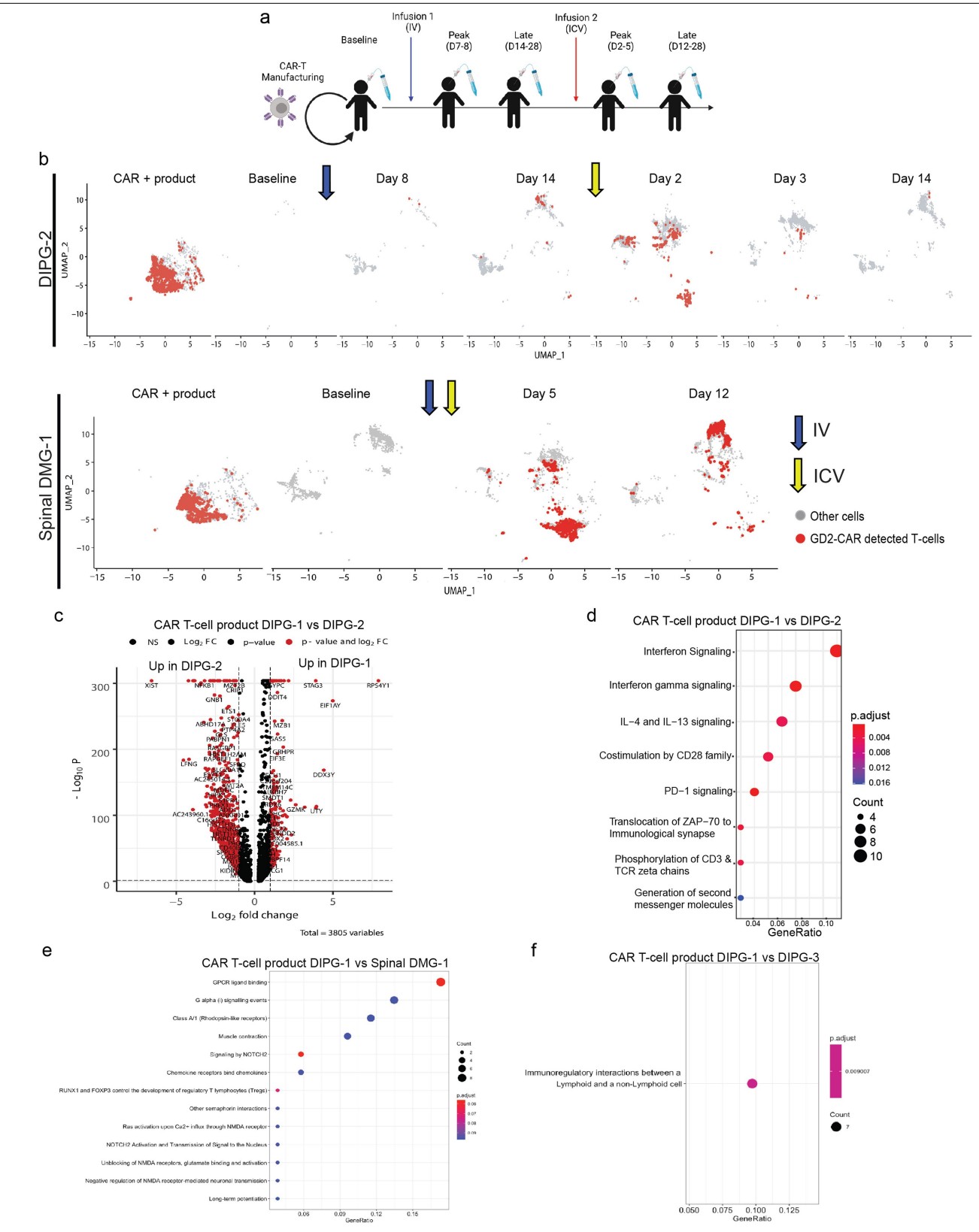

**Extended Data Fig. 6** | See next page for caption.

**Extended Data Fig. 6 | Single cell RNA-sequencing of GD2-CAR T cells in CSF and from manufactured product. a**, Schematic of single cell RNA sequencing (scRNA-seq) via 10X Genomics platform, conducted on sorted GD2-CAR+ and CAR- infusion products, as well as CSF samples at indicated timepoints throughout treatment. 65,598 single cells were sequenced, including 20,000 single cells from the CAR+ fraction of patient products and 25,598 cells obtained from patient CSF (shown in Fig. 4). Schematic created with BioRender. com. **b**, GD2-CAR+ T cells were identified in post-treatment CSF samples of DIPG-2 and Spinal DMG-1 following GD2-CAR T cell administration. T cells with detected GD2-CAR expression are represented as red dots. A single UMAP was generated with flow-sorted GD2-CAR T cell product and cells from the CSF then individually visually represented based on day of the sample. Note alterations in GD2-CAR T cell profile over time. CSF studies were not obtained for Spinal DMG-1 following IV infusion. CSF studies were also limited in DIPG-3: Following ICV GD2-CAR T cell administration, DIPG-3 did not require CSF drainage during the period of peak inflammation and given her young age, elective CSF collection was more limited. Therefore, the only post-ICV timepoint for CSF collection from DIPG-3 was Day 14, at which point there were not enough cells in CSF to conduct scRNAseq. Blue arrow=IV infusion, Yellow arrow=ICV infusion. **c**, Volcano plot representing CAR T cell product of DIPG-1 compared to DIPG-2. **d**–**f**, Gene ontology term analysis of the most significantly enriched pathways in DIPG-1CAR+ product relative to DIPG-2 (**d**), Spinal DMG-1 (**e**), and DIPG-3 (**f**) CAR+ product prior to CAR T cell administration. The y-axis shows the enhanced gene ontology terms, while the x-axis ("GeneRatio") corresponds to the overlap between the up-regulated genes and genes associated with the given gene ontology terms in the dataset. Color of the circle corresponds to the p-value significance of the pathway enrichment, relative to all other genes in the dataset. Size of the circle corresponds to the relative number of matching up-regulated genes in the given gene ontology term.

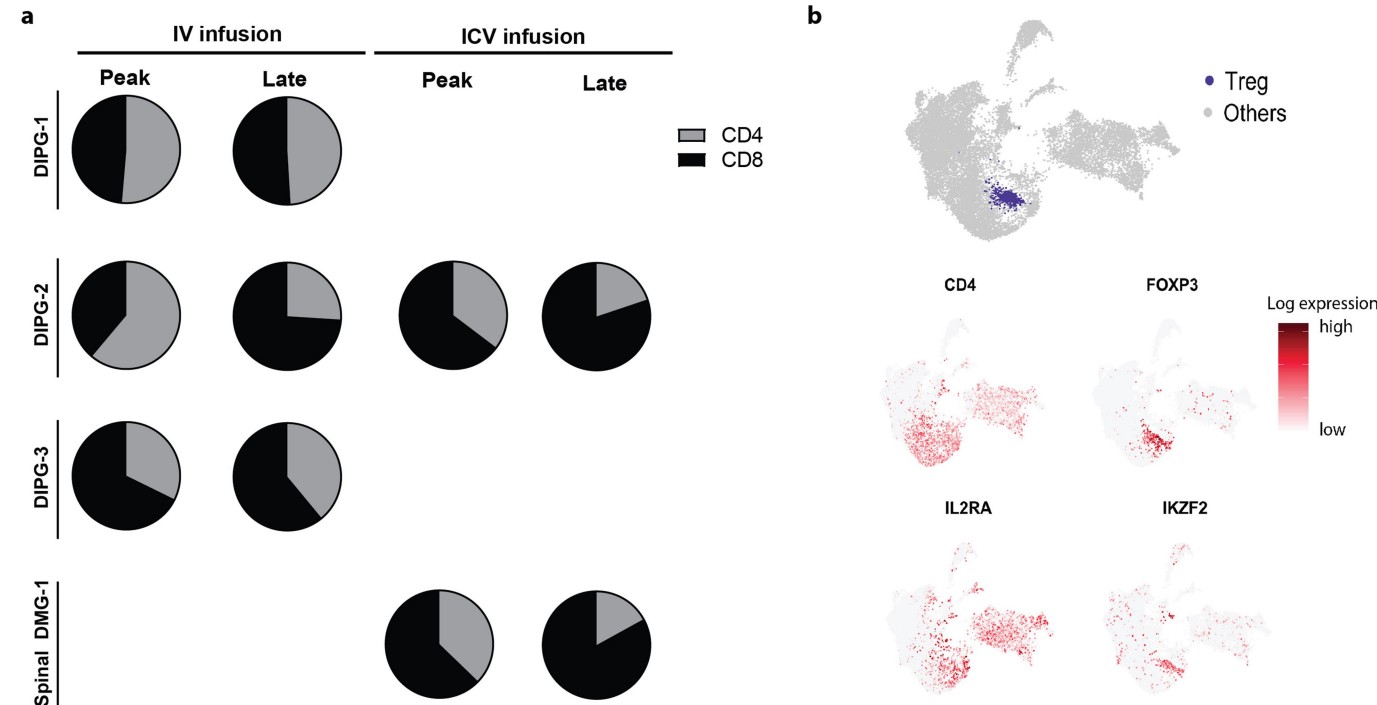

**Extended Data Fig. 7 | CD4:CD8 Ratios of T cells throughout treatment and gene ontology term analysis of CAR T cell products prior to CAR T cell treatment. a**, Pie charts represent CSF T cell CD4:CD8 ratios from patients at different timepoints following CAR T cell administration. **b**, Top panel: CSF T cell UMAP projections highlighting Tregs in scRNA-seq data. The Treg cell cluster is highlighted in purple, while all other T cells are colored grey. Bottom panel: UMAP projections highlighting Tregs in scRNA-seq data by expressions of canonical marker genes of Tregs (CD4, FOXP3, CD25, and IKZF2).

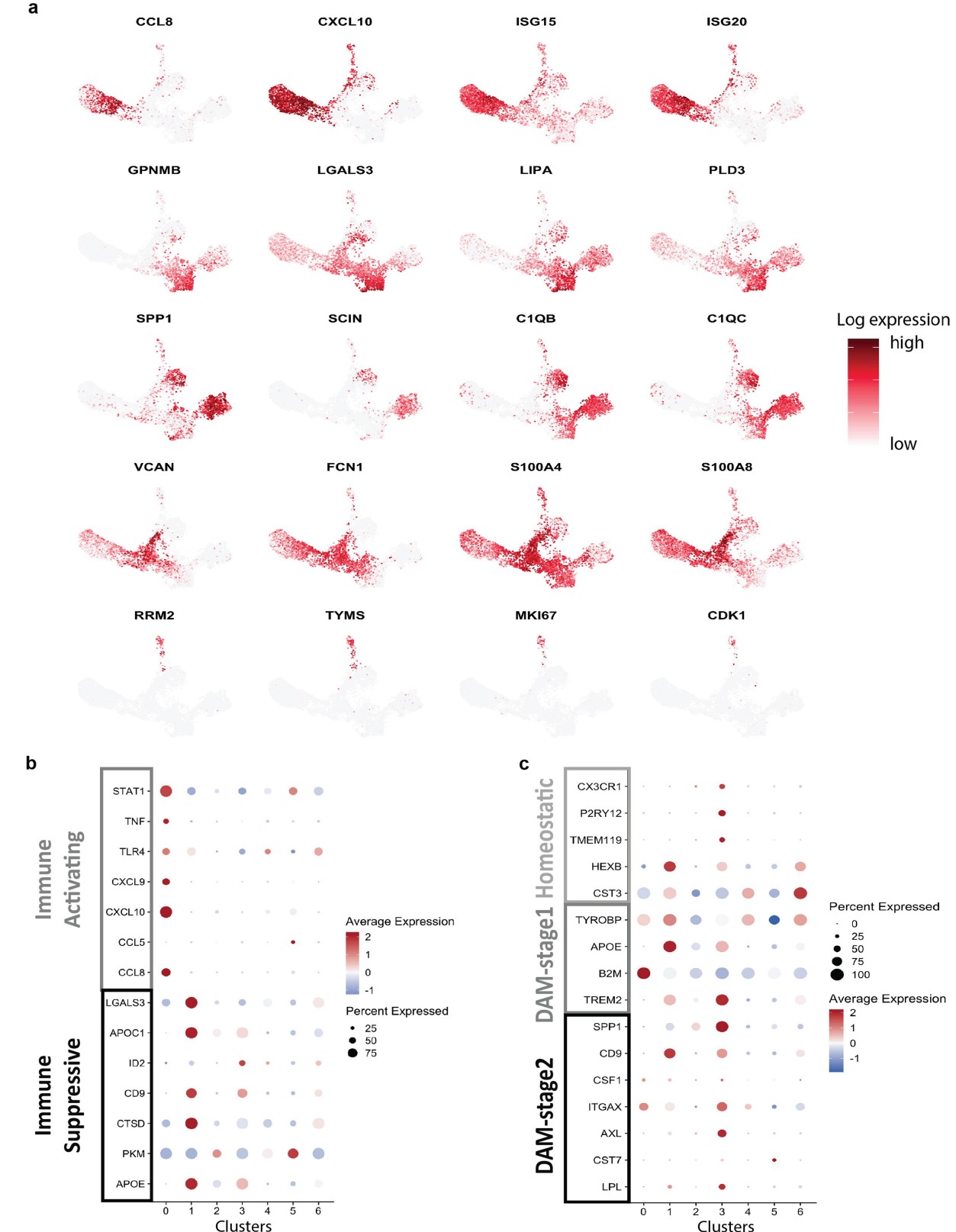

**Extended Data Fig. 8 | Expression of cell cluster-specific marker genes and myeloid signature genes. a**, CSF myeloid cell UMAP projections highlighting myeloid cell clusters in scRNA-seq data by top differentially expressed genes of each cell cluster. **b**, **c**, Expression of myeloid transcriptional signature genes (y-axis) of identified myeloid cell clusters (x-axis). Dot sizes represent the percentage of cells expressing the gene in the given cluster. Color scale shows scaled average expression. Genes of immune activating and suppressive (**b**), homeostatic, and DAM stage 1 and 2 signatures (**c**) are plotted.

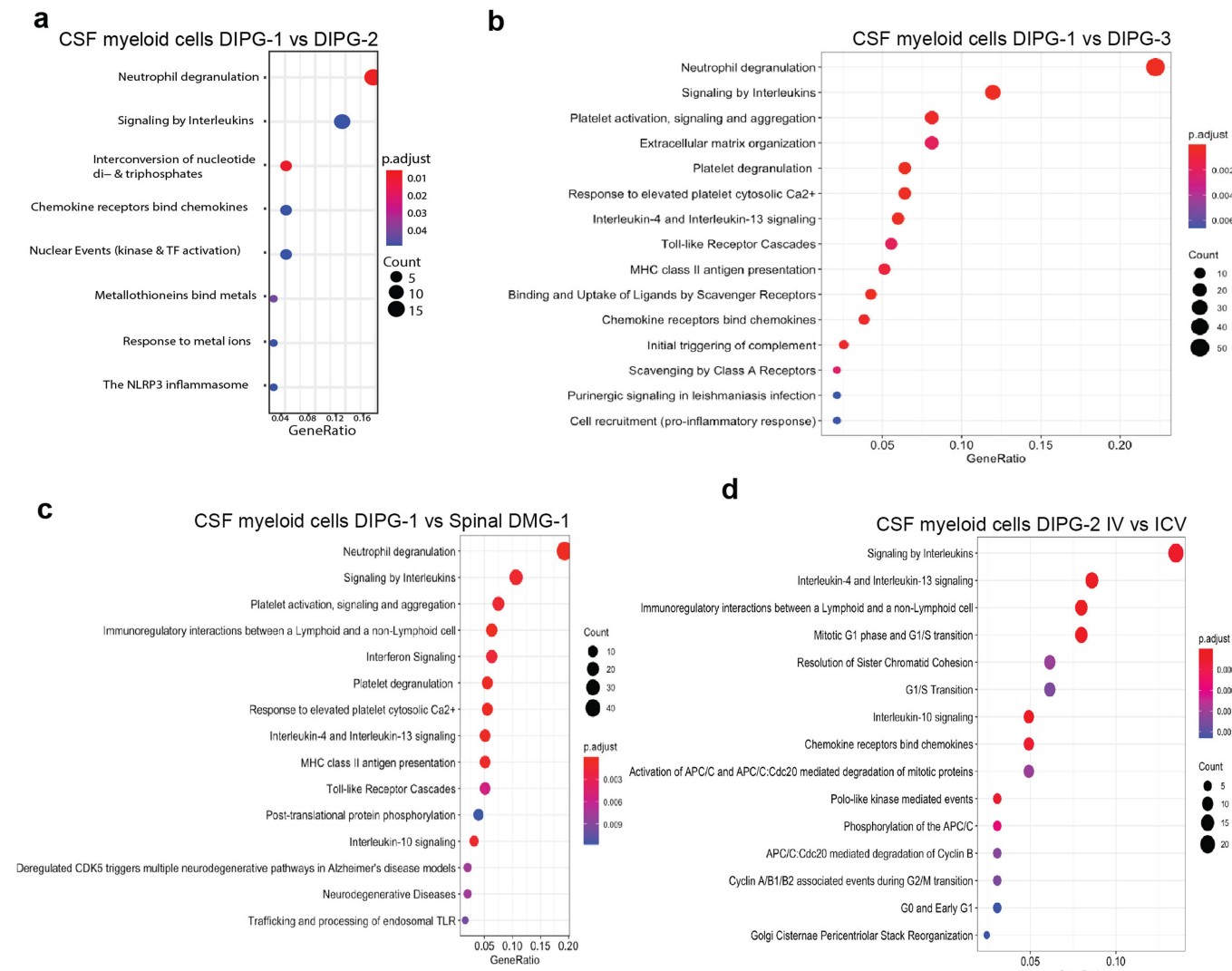

**Extended Data Fig. 9 | Gene ontology term analysis of myeloid cells from CSF following GD2-CAR T cell treatment. a–c**, Gene ontology term analysis demonstrates the most significantly activated pathways in the myeloid fraction of DIPG-1 compared to DIPG-2 (**a**), DIPG-3 (**b**), and Spinal DMG-1 (**c**). **d**, Gene ontology term analysis demonstrates the most significantly enriched pathways in the myeloid fraction of DIPG-2 CSF sample on Day 8 of IV CAR T administration relative to the myeloid fraction of DIPG-2 CSF sample on Day 2 of ICV CAR T cell administration (time points of peak inflammation following IV or ICV administration for this patient). The y-axis shows the enhanced gene ontology terms, while the x-axis ("GeneRatio") corresponds to the overlap between the up-regulated genes and genes associated with the given gene ontology term in the dataset. Color of the circle corresponds to the p-value significance of the pathway enrichment, relative to all other genes in the dataset. Size of the circle corresponds to the relative number of matching up-regulated genes in the given gene ontology term. Only myeloid cells in CSF are included in this analysis.

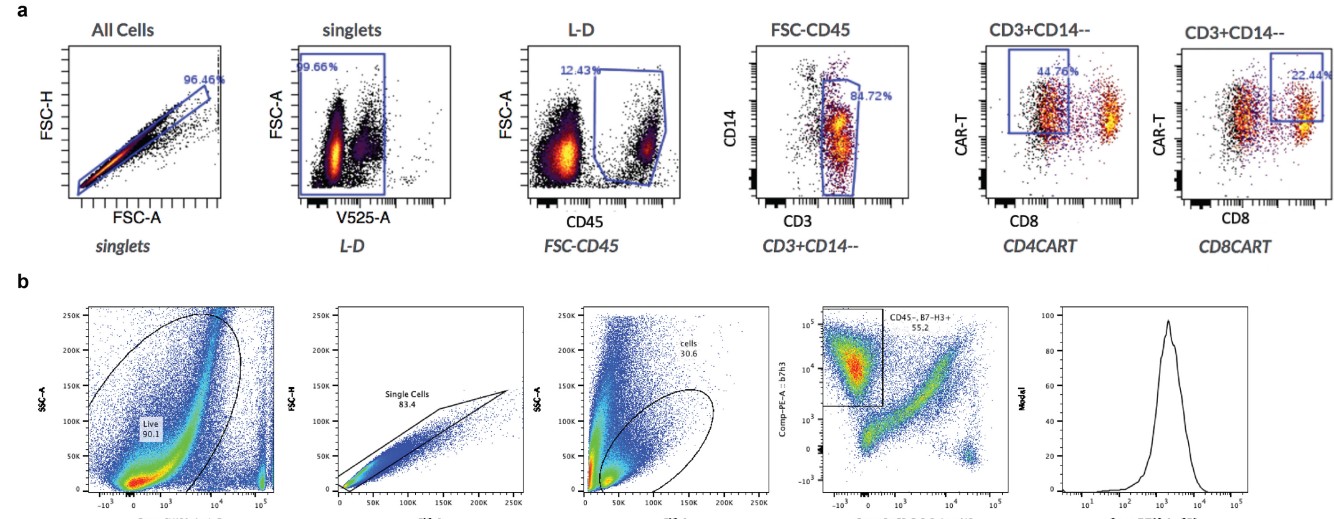

**Extended Data Fig. 10 | Representative gating for flow cytometry of patient samples. a**, Representative gating of a patient CSF sample by flow cytometry (identical gating was used for PBMC obtained from blood). **b**, Gating of tumour cells by flow cytometry from temporal lobe resection tissue from Spinal DMG-1.

# Reporting Summary

## Statistics

For all statistical analyses, confirm that the following items are present in the figure legend, table legend, main text, or Methods section.

| n/a | Confirmed | |
|---|---|---|
| ☐ | ☒ | The exact sample size (*n*) for each experimental group/condition, given as a discrete number and unit of measurement |
| ☐ | ☒ | A statement on whether measurements were taken from distinct samples or whether the same sample was measured repeatedly |
| ☐ | ☒ | The statistical test(s) used AND whether they are one- or two-sided *Only common tests should be described solely by name; describe more complex techniques in the Methods section.* |
| ☐ | ☒ | A description of all covariates tested |
| ☒ | ☐ | A description of any assumptions or corrections, such as tests of normality and adjustment for multiple comparisons |
| ☒ | ☐ | A full description of the statistical parameters including central tendency (e.g. means) or other basic estimates (e.g. regression coefficient) AND variation (e.g. standard deviation) or associated estimates of uncertainty (e.g. confidence intervals) |
| ☒ | ☐ | For null hypothesis testing, the test statistic (e.g. *F*, *t*, *r*) with confidence intervals, effect sizes, degrees of freedom and *P* value noted *Give P values as exact values whenever suitable.* |
| ☒ | ☐ | For Bayesian analysis, information on the choice of priors and Markov chain Monte Carlo settings |
| ☒ | ☐ | For hierarchical and complex designs, identification of the appropriate level for tests and full reporting of outcomes |
| ☒ | ☐ | Estimates of effect sizes (e.g. Cohen's *d*, Pearson's *r*), indicating how they were calculated |

*Our web collection on statistics for biologists contains articles on many of the points above.*

## Software and code

Policy information about availability of computer code

| Data collection | N/A |
|---|---|
| Data analysis | Analysis of single cell RNA sequencing data was performed with referenced custom R scripts. Data analysis was performed in Graphpad Prism and Microsoft Excel. |

For manuscripts utilizing custom algorithms or software that are central to the research but not yet described in published literature, software must be made available to editors and reviewers. We strongly encourage code deposition in a community repository (e.g. GitHub). See the Nature Portfolio guidelines for submitting code & software for further information.

## Data

Policy information about availability of data

All manuscripts must include a data availability statement. This statement should provide the following information, where applicable:

- Accession codes, unique identifiers, or web links for publicly available datasets
- A description of any restrictions on data availability
- For clinical datasets or third party data, please ensure that the statement adheres to our policy

All raw data are provided in the source data file. The single cell RNA-sequencing data has been uploaded to GEO (GSE186802).

# Field-specific reporting

Please select the one below that is the best fit for your research. If you are not sure, read the appropriate sections before making your selection.

☒ Life sciences ☐ Behavioural & social sciences ☐ Ecological, evolutionary & environmental sciences

For a reference copy of the document with all sections, see nature.com/documents/nr-reporting-summary-flat.pdf

# Life sciences study design

All studies must disclose on these points even when the disclosure is negative.

| | |
|---|---|
| Sample size | Four patients were enrolled on the first dose level of this clinical trial. |
| Data exclusions | No data was excluded from the analysis. |
| Replication | ctDNA, and RT-PCR for CAR transgene were performed in triplicate, Cytokine analyses were performed in duplicate. |
| Randomization | No randomization was performed in this Phase 1 clinical trial. |
| Blinding | No blinding was performed in this Phase 1 clinical trial. |

# Reporting for specific materials, systems and methods

We require information from authors about some types of materials, experimental systems and methods used in many studies. Here, indicate whether each material, system or method listed is relevant to your study. If you are not sure if a list item applies to your research, read the appropriate section before selecting a response.

## Materials & experimental systems

| n/a | Involved in the study |
|---|---|
| ☐ | ☒ Antibodies |
| ☒ | ☐ Eukaryotic cell lines |
| ☒ | ☐ Palaeontology and archaeology |
| ☒ | ☐ Animals and other organisms |
| ☐ | ☒ Human research participants |
| ☐ | ☒ Clinical data |
| ☒ | ☐ Dual use research of concern |

## Methods

| n/a | Involved in the study |
|---|---|
| ☒ | ☐ ChIP-seq |
| ☐ | ☒ Flow cytometry |
| ☐ | ☒ MRI-based neuroimaging |

# Antibodies

| | |
|---|---|
| Antibodies used | CD3 FITC UCHT1 BioLegend 300406<br>CD8 PerCP Cy5.5 SK1 BD Pharmingen 565310<br>CD45 BV785 2D1 BioLegend 368528<br>CD4 BV711 RPA-T4 BioLegend 300558<br>CD95 BV650 DX2 BioLegend 305624<br>CD39 Bv605 A1 BioLegend 328236<br>Cell viability BV510 N/A Invitrogen L-34965<br>CD57 BV421 NK-1 BDBiosciences 563896<br>CCR7 BUV805 2L1A BDBiosciences 749673<br>CD45RA Alx700 HI100 BioLegend 304120<br>GD2CAR DyLight650 1A7 Custom<br>CD14 PE-Cy7 63D3 BioLegend 367112<br>CD11b APC-Cy7 ICRF44 BioLegend 301352<br>CD33 PE-Dazzle WM53 BIolegend 303432<br>GD2 PE 14G2A BioLegend 357304<br>CD4 BUV395 SK3 BD Biosciences 563550<br>CD8 BUV795 SK1 BD Biosciences 564912<br>CD45 PerCP-Cy5.5 HI30 eBioscience 45-0459-41<br>GD2 BV510 14g2a BioLegend 357316<br>B7-H3 PE AF R&D FAP1027P<br>CD14 PE-Cy7 63D3 BioLegend 367112<br>CD11b APC-Cy7 ICRF44 BioLegend 301352<br>Cell viability DAPI N/A ThermoFisher Scientific 62247<br>GD2CAR DyLight650 1A7 Custom |

| Validation | All antibodies were validated as per the manufacturer except 1A7. In the case of 1A7, the anti-GD2 CAR idiotype antibody, untransduced T cells were used as a biologic control. |
|---|---|
| | For RNAscope analysis of the GD2 CAR construct, validation was performed using a positive control (cultured GD2-CAR T-cells) and a negative control (autopsy brain tissue sample from an untreated patient). |

## Human research participants

Policy information about studies involving human research participants

| Population characteristics | Beginning in June 2020, four subjects were enrolled on DL1 (1e6 GD2-CAR T/kg; 3 DIPG, 1 spinal cord DMG; ages 5-25; 1M/3F). Results are reported here with a data cut of March 2021. |
|---|---|
| Recruitment | Patients on this Phase 1 clinical trial were recruited through physician and self-referral. |
| Ethics oversight | The Stanford University IRB approved this clinical study. |

Note that full information on the approval of the study protocol must also be provided in the manuscript.

## Clinical data

Policy information about clinical studies

All manuscripts should comply with the ICMJE guidelines for publication of clinical research and a completed CONSORT checklist must be included with all submissions.

| Clinical trial registration | NCT04196413 |
|---|---|
| Study protocol | Study protocol provided with ammendments. |
| Data collection | June 2020- March 2021 |
| Outcomes | Primary objectives assessed: feasibility of manufacturing, safety and tolerability, and identifying the maximally tolerated dose or recommended phase II dose. Assessment of clinical activity is the secondary objective and identifying correlative biomarkers of response is an exploratory objective. |

## Flow Cytometry

### Plots

Confirm that:

☒ The axis labels state the marker and fluorochrome used (e.g. CD4-FITC).

☒ The axis scales are clearly visible. Include numbers along axes only for bottom left plot of group (a 'group' is an analysis of identical markers).

☒ All plots are contour plots with outliers or pseudocolor plots.

☒ A numerical value for number of cells or percentage (with statistics) is provided.

### Methodology

| Sample preparation | PBMC were isolated from fresh whole blood by gradient centrifugation on ficoll (Ficoll paque Plus, GE Healthcare, SigmaAldrich). Two to five million PBMC were stained with fixable Live/Dead aqua (Invitrogen) amine-reactive viability stain. Cells were then preincubated with Fc block (trustain, Biolegend) for 5 min, then stained at room temperature with the following fluorochrome conjugated mAb in an 15-color, 17-parameter staining combination. |
|---|---|
| Instrument | LSR (BD BioSciences) |
| Software | Analysis was performed in FlowJo |
| Cell population abundance | At least, 106 cells were acquired unless restricted by the number of cells isolated from 8 ml of whole blood or when acquiring CSF isolated cells. The assay limit of detection for cells calculated as 1 in 104 of total acquired PBMCs. |
| Gating strategy | Tumor resection material: Viable cells->singlets->cells (FSC/SSC)->CD45-/B7-H3+. GD2 expression compared to FMO control |
| | CAR T-cells in CSF and PBMC: Singlets->viable cells->CD45+->CD14-, CD3+ -> CD4 or CD8 CAR positivity gated based on control PBMC. |

☒ Tick this box to confirm that a figure exemplifying the gating strategy is provided in the Supplementary Information.

# Magnetic resonance imaging

## Experimental design

| | |
|---|---|
| Design type | Clinical studies |
| Design specifications | Clinical studies |
| Behavioral performance measures | NA |

## Acquisition

| | |
|---|---|
| Imaging type(s) | MRI brain and spine (clinical protocols) |
| Field strength | 3T |
| Sequence & imaging parameters | T2 sequences shown |
| Area of acquisition | Brain and Spine |

Diffusion MRI ☐ Used ☒ Not used

## Preprocessing

| | |
|---|---|
| Preprocessing software | NA |
| Normalization | NA |
| Normalization template | NA |
| Noise and artifact removal | NA |
| Volume censoring | NA |

## Statistical modeling & inference

| | |
|---|---|
| Model type and settings | NA |
| Effect(s) tested | NA |

Specify type of analysis: ☒ Whole brain ☐ ROI-based ☐ Both

| | |
|---|---|
| Statistic type for inference (See Eklund et al. 2016) | NA |
| Correction | NA |

## Models & analysis

| n/a | Involved in the study |
|---|---|
| ☒ | ☐ Functional and/or effective connectivity |
| ☒ | ☐ Graph analysis |
| ☒ | ☐ Multivariate modeling or predictive analysis |

