## [Peer Review File · Nature]

Manuscript Title: GD2-CAR T-cell therapy for H3K27M-mutated diffuse midline gliomas

Reviewer Comments & Author Rebuttals

Reviewer Reports on the Initial Version:

Referee #1 (Remarks to the Author):

Majzner et al describe the clinical results of the first 4 patients treated on a phase I clinical study of GD2-targeted CAR-T cell therapy for spinal/pontine diffuse midline glioma (DMG). They show clinical and radiological evidence of anti-tumor activity in 3 of 4 patients treated. The clinical data are complemented with correlative studies which begin to elucidate the interplay between CAR-T cells, tumor and its microenvironment and which may provide insights into how to further refine this treatment approach.

DMG/DIPG is a devastating disease with no therapeutic advance for decades, hence the reported first indication of anti-tumor activity in this disease is highly significant and represents a true breakthrough. In addition, while CAR-T cell therapy has made huge progress in hematological malignancies its role in solid tumors is as of yet unclear - presented data which show that CAR-T cell therapy has activity in a difficult to treat tumor like DMG is again very significant. The authors are to be congratulated for initiating this challenging but highly important clinical trial; the emerging results will be of great interest to a broad readership not in the least the large field of brain tumor/pathology research and that of immunotherapies for solid tumors.

My comments are centered around showing a more complete clinical picture of the patients treated on this study and objective measures of radiological responses.

Main comments:

* It is well recognized that quantification of tumor volume for DMG/DIPG is challenging due to the infiltrative growth of these tumors. Both T2 and FLAIR sequences have been used to measure tumor diameter in the context of DMG [TM Cooney et al, Lancet Oncol 2020, PMID:32502459]. A short description of how tumor volumes are measured is provided. While for 2 patients graphs showing tumor volume over time are provided, this information is not available for all patients at all time points. To comprehensively show radiological responses and changes over time:

- please include the measured tumor volumes for all 4 patients at all time points so that it is clear which tumor volumes are used to calculate the percentage change. To ensure the disease course is clearly captured please indicate which baseline is used for this calculation as for example DIPG subject 2 has significant disease progression prior to the second CAR-T infusion
- please include in the figure panel itself (i.e. bottom corner of each image) or in the figure legend the volumetric tumor measurement at each time point for which a representative MRI image is shown
- in extended Figure 1d which shows tumor volumes for DIPG subject 2 please include all time points i.e. also d+7 and d+21
- to provide a complete overview of the clinical course for DIPG subject 3 both responding and non-responding areas of tumor should be shown in the main figure. In Figure 1i please include images showing worsening of radiological appearance to illustrate a mixed response.

* As done for DIPG subject 1, please include for all patients what the eventual clinical outcomes were

* Line 299-302: It is unusual not to be able to detect CAR-T cells by flow cytometry. As systemically there will be no GD2 exposure, internalization of the receptor is unlikely. Is there any alternative explanation for discrepancy in CAR-T cell detection by PRC as compared to by flow cytometry?

Minor comments:

- * Text line 198: 'a 27% reduction in tumor volume compared to his pre-infusion MRI'. Here please clarify here 'pre-infusion' refers to the baseline prior to the second CAR-T cell infusion
- * Text line 247: please add the route of administration
- * Figure 1c: GD2 staining looks very diffuse whereas it should be restricted to the membrane. It will be useful to also have a negative control, i.e. a tissue without GD2 expression included in extended data. Please include the method of GD2 staining in the material and methods section.
- * Figure 1d: doing qPCR on bulk sample is helpful, but would be extremely useful to see which proportion of infiltrating T cells are CAR-T either by staining for transcripts or CAR (anti-idiotype). This is important as the real time PCR signal seems low given the amount of lymphocytic infiltrate in the biopsy/autopsy tissue
- * Figure 2f: cfDNA free may show an initial increase in the context of tumor lysis followed by a decrease reflecting tumor load [ref 14]. Time points of detected cDNA in CSF post IV and post ICV administration are different i.e. an increase post ICV CAR-T administration is seen on day+3 and day+6 post ICV (DIPG-2) and day+5 (spinal DMG), however earliest time point post IV is day+9 and day+28 which may mean that an initial peak due to tumor lysis is missed. CSF at additional time points may not be available, however please include this caveat when referring to these results (line 290-293).
- * Figure 3: data is this figure is clearly presented allowing an overview of key inflammatory markers elevated as well as changes over time. However, as data is shown as change over baseline it does not show concentrations of inflammatory markers measured - please provide this information in an extended data table
- * Figure 4d: The results are described as 'tracking GD2+ CAR T-cells in CSF after IV and ICV administration revealed more CAR T-cells detected in CSF at times of peak inflammation, and higher CAR T-cells in CSF after ICV administration than after IV administration for DIPG Subject 2 (Figure 4d)'. DIPG subject 3 similarly received both an IV and ICV CAR-T infusion: what did the comparison of CAR-T as detected in CSF for this patient show?
- * Extended data Figure 1b - please include normal cortex comparator similar to as shown in Figure 1c
- * Extended data figure 3d: please explain what Sup_1, Sup_10 etc refer to.
- * Extended Table 1: please include time off treatment when treatment on this phase 1 clinical study started

Referee #2 (Remarks to the Author):

Title: "GD2-CAR T-cell therapy for H3K27M-mutated diffuse midline gliomas" by Professor Monje and co-authors (manuscript number 2021-07-11811)

Summary: Majzner, Ramakrishna et al. describe findings from their phase 1 clinical trial (NCT04196413) to treat H3K27M-mutated diffuse midline gliomas and demonstrate clinical improvement following CAR T cell treatment in three of four patients with manageable toxicity profiles. In addition, they perform multiple lines of molecular profiling and provide valuable insights for heterogeneity in response. Overall, the authors demonstrate very promising tolerability and response results for the use of CAR T cells targeting the disialoganglioside GD2 for H3K27M diffuse midline gliomas.

Comments: This manuscript is well-written, clear and is beautifully illustrated. Dr. Monje is a world leader in this field and this exceptional study (in the true spirit of a physician scientist) is based on a previous landmark preclinical paper in H3K27M models. The current study is superlative and is a major advance in not only understanding the biology of these tumors but also provides a promising treatment for this fatal childhood cancer. They provide great detail in their characterization of toxicities and their responses, which will serve as valuable data for future immune-driven approaches to treating brainstem-associated and other tumors. This exceptional

paper will be of great interest to the pediatric brain tumor community, and the correlative supporting data will serve as a valuable example of translational research for rare tumors. The following minor comments are noted to improve the clarity and main message of this paper:

Minor comments:

1) The single-cell RNA-seq data provide incredible, valuable insight into potential predictive factors about engineered CAR T cells and their likelihood of producing a clinical response. As illustrated in Ext. Data Table 2, there are considerable differences in cell manufacturing for DIPG 1 compared to the other tumors. Do the authors have any metrics on the initial autologous peripheral blood mononuclear cell populations that may shed light on the differences in the resulting CAR T % and CD4+/CD8+ breakdown?

2) The authors primarily compare DIPG 1 and DIPG 2 populations from either the starting product or from CSF myeloid cells to identify pathways that may explain the differences in clinical response. Do any of the findings highlighted in Fig. 4e and f persist in comparisons to the other two tumors? If not, are there other changes that may explain the differences in clinical response?

3) Considering the differences in the starting population of DIPG 1 CAR T cells compared to the other tumors (Ext. Data Table 2), can the authors provide information (if available) on the percentage of GD-2 CAR T cells that make up the CSF population for later time points in DIPG 1 as done for DIPG 2 and Spinal DMG. Additionally characterization of the CD4/8 ratio of these CSF GD-2 CAR T cells in the various patients could shed light on potential relationships with clinical response.

4) The authors describe a population of cells in Ext. Data Fig5d and e that is unique to DIPG 1. Does this population remain independent if myeloid cells from the other tumors are incorporated? What genes drive this cluster?

5) Can the authors provide clarity on the figure legend for Fig 1e. Does "tumor volume" include both spinal and brain disease?

6) It is very interesting that the DIPG 1 GD2 CAR DNA levels do not appear to drop as they do in the other patients' samples. Could this be related to the clinical response differences?

Referee #3 (Remarks to the Author):

The authors present a series of four patients who have been treated on a novel and exciting clinical trial in which they were treated with intra-CNS immunotherapy for DIPG.

The plan and execution of the clinical trial are novel, innovative, and likely first in class for the globe. This is an exciting avenue for the future treatment of brain tumors. The documentation of the cases is excellent, and there is an amount of correlative science to accompany the clinical data.

Unfortunately, this is a series of only four patients, and while the preliminary results are interesting, I would suggest that the authors be encouraged to resubmit once they have a larger cohort, as this cohort is too small to draw any conclusions, and will be of limited interest at this point to the broad readership of Nature.

I would be highly in favor of inviting the authors to resubmit when their cohort is larger, and when more of the patients have been observed to endpoint.

Author Rebuttals to Initial Comments:

We were delighted to see the positive comments and are grateful for the thoughtful and helpful suggestions that have improved the manuscript. We have addressed all referee comments, detailed below point-by-point.

Referees' comments:

Referee #1 (Remarks to the Author):

Majzner et al describe the clinical results of the first 4 patients treated on a phase I clinical study of GD2-targeted CAR-T cell therapy for spinal/pontine diffuse midline glioma (DMG). They show clinical and radiological evidence of anti-tumor activity in 3 of 4 patients treated. The clinical data are complemented with correlative studies which begin to elucidate the interplay between CAR-T cells, tumor and its microenvironment and which may provide insights into how to further refine this treatment approach.

DMG/DIPG is a devastating disease with no therapeutic advance for decades, hence the reported first indication of anti-tumor activity in this disease is highly significant and represents a true breakthrough. In addition, while CAR-T cell therapy has made huge progress in hematological malignancies its role in solid tumors is as of yet unclear - presented data which show that CAR-T cell therapy has activity in a difficult to treat tumor like DMG is again very significant. The authors are to be congratulated for initiating this challenging but highly important clinical trial; the emerging results will be of great interest to a broad readership not in the least the large field of brain tumor/pathology research and that of immunotherapies for solid tumors.

We thank the Referee for their detailed review of this manuscript and for the positive and supportive comments.

My comments are centered around showing a more complete clinical picture of the patients treated on this study and objective measures of radiological responses.

We appreciate the comments and have revised the paper according to these helpful suggestions.

Main comments:

* It is well recognized that quantification of tumor volume for DMG/DIPG is challenging due to the infiltrative growth of these tumors. Both T2 and FLAIR sequences have been used to measure tumor diameter in the context of DMG [TM Cooney et al, Lancet Oncol 2020, PMID:32502459]. A short description of how tumor volumes are measured is provided. While for 2 patients graphs showing tumor volume over time are provided, this information is not available for all patients at all time points. To comprehensively show radiological responses and changes over time:

- please include the measured tumor volumes for all 4 patients at all time points so that it is clear which tumor volumes are used to calculate the percentage change. To ensure the disease course is clearly captured please indicate which baseline is used for this calculation as for example DIPG subject 2 has significant disease progression prior to the second CAR-T infusion

- please include in the figure panel itself (i.e. bottom corner of each image) or in the figure legend the volumetric tumor measurement at each time point for which a representative MRI image is shown
- in extended Figure 1d which shows tumor volumes for DIPG subject 2 please include all time points i.e. also d+7 and d+21
- to provide a complete overview of the clinical course for DIPG subject 3 both responding and non-responding areas of tumor should be shown in the main figure. In Figure 1i please include images showing worsening of radiological appearance to illustrate a mixed response.

We have now included tumor volumetric measurements for all 4 patients (Figure 2 and Extended Figure 1). We have also added additional tumor volume timepoints (D+7 and D+21 (Extended Figure 1d)) for DIPG subject 2 and have added additional images of DIPG subject 3 better illustrating the responding sites and non-responding sites (Figure 2). We have clarified the baseline timepoint used to assess a change in tumor volume, and have added the time points of each MRI image in the legends or clearly indicated on the figure itself. We thank the Referee for these helpful suggestions, which improve the paper.

* As done for DIPG subject 1, please include for all patients what the eventual clinical outcomes were

We appreciate this Referee's comment to include the eventual clinical outcomes of all patients. We have added these outcomes in each patient section as follows:

- DIPG 2: "After the data cutoff of March 2021, this subject went on to receive 3 more ICV infusions (5 infusions total). Prior to the planned 6th infusion, he died due to an intratumoral hemorrhage in a known area of intratumoral vascular anomaly. Such intratumoral hemorrhages are relatively common in DIPG and risk increases with time from diagnosis (Bronscier et al., 2006). He survived 26 months from diagnosis, 12 months after beginning to exhibit radiographic and clinical progression, and 10 months after his first infusion." (Lines 199-204)
- DIPG 3: "After the data cutoff of March 2021, this subject received one more ICV infusion but her tumor continued to progress. She died from tumor progression 20 months after diagnosis, 4 months after beginning to exhibit radiographic progression, and 7 months after first infusion." (Line 242-245)
- Spinal DMG 1: "After the data cutoff of March 2021, she survived 20 months from diagnosis, 14 months from the beginning of clinical and radiographic progression and 11 months after the first infusion." (Line 296-298)

* Line 299-302: It is unusual not to be able to detect CAR-T cells by flow cytometry. As systemically there will be no GD2 exposure, internalization of the receptor is unlikely. Is there any alternative explanation for discrepancy in CAR-T cell detection by PCR as compared to by flow cytometry?

We agree that the discrepancy of CAR-T cell detection by PCR compared to flow is unusual. These findings could indicate GD2 CAR internalization – perhaps in the context of activation in the CNS and trafficking back to the systemic circulation through meningeal lymphatics - but also

could reflect cells with low copy number of the gene resulting in GD2-CAR expression below the limit of detection by flow or silencing of the transgene. We are currently working to understand this discrepancy in the laboratory. Since PCR is the gold-standard in tracking CAR persistence, we have used this measure to reflect the truest representation of the trajectory of CAR expansion, contraction, and persistence.

Minor comments:

* Text line 198: 'a 27% reduction in tumor volume compared to his pre-infusion MRI'. Here please clarify here 'pre-infusion' refers to the baseline prior to the second CAR-T cell infusion. Thank you for this helpful suggestion, we have now edited the text to clarify that “pre-infusion” refers to the baseline prior to the second infusion (Line 196).

* Text line 247: please add the route of administration

Thank you for identifying this point requiring clarification. We have edited to state that this dose was administered intravenously.

* Figure 1c: GD2 staining looks very diffuse whereas it should be restricted to the membrane. It will be useful to also have a negative control, i.e. a tissue without GD2 expression included in extended data. Please include the method of GD2 staining in the material and methods section. We appreciate this helpful suggestion. We have now added human muscle tissue as a negative control (Extended Data Figure 1e and shown below). We have also included more details of GD2 staining in the materials and methods section (Lines 1087 - 1104). The Referee is correct that the staining pattern for GD2 in DIPG is quite diffuse, and this is consistent our past experience with GD2 immunohistochemistry in DIPG tumors (Mount et al., 2018). Similar staining patterns are seen in Marconi et al, Journal of Neuroimmunology, 2005.

Extended Data Figure 1a: Normal human muscle tissue stained for GD2 as negative control for GD2 antigen staining.

* Figure 1d: doing qPCR on bulk sample is helpful, but would be extremely useful to see which proportion of infiltrating T cells are CAR-T either by staining for transcripts or CAR (anti-idiotype). This is important as the real time PCR signal seems low given the amount of lymphocytic infiltrate in the biopsy/autopsy tissue

Thank you for this very helpful comment. The anti-idiotype antibody does not work well by immunohistochemistry, so we developed an RNAscope probe for single molecule RNA *in situ* visualization. The RNAscope probe was validated in GD2 CAR T-cells in culture (positive control) and in tissue from postmortem tissue from a patient not treated with CAR T-cells (negative control) (Extended Data Figure 1d and shown below). RNAscope analysis identified

CAR-T cells in tumor tissue, but not in cortex, from DIPG Subject 1 (Figure 1c and shown below). This illustrates that a subset of T cells in the tumor tissue represent CAR T-cells.

LEFT (Figure 1c, third column from left): GD2-CAR mRNA expression detected by RNAscope identified GD2-CAR T-cells in tumor but not in areas of normal brain. GD2-CAR+ cells = pink, hematoxylin counterstain for all cells = blue.

RIGHT (Extended Data Figure 1d): Positive control GD2-CAR T-cells in tissue culture and negative control cortex from a DIPG patient not treated with CAR T-cells were used to validate the RNAscope probe.

* Figure 2f: cfDNA free may show an initial increase in the context of tumor lysis followed by a decrease reflecting tumor load [ref 14]. Time points of detected cDNA in CSF post IV and post ICV administration are different i.e. an increase post ICV CAR-T administration is seen on day+3 and day+6 post ICV (DIPG-2) and day+5 (spinal DMG), however earliest time point post IV is day+9 and day+28 which may mean that an initial peak due to tumor lysis is missed. CSF at additional time points may not be available, however please include this caveat when referring to these results (line 290-293).

We agree that there may be timepoints earlier in the treatment course that may better represent the initial peak due to tumor lysis. Unfortunately, we do not have CSF samples at these earlier timepoints to assess for cfDNA, as samples were collected for cfDNA analysis at pre-defined timepoints or when clinically indicated. We have added this caveat to the manuscript, stating: “Levels of ctDNA may have peaked at timepoints other than those captured, and additional samples could help further elucidate the kinetics of tumor cell killing in future studies.” (Line 313-315).

* Figure 3: data in this figure is clearly presented allowing an overview of key inflammatory markers elevated as well as changes over time. However, as data is shown as change over baseline it does not show concentrations of inflammatory markers measured - please provide this information in an extended data table

We agree with the Referee that concentrations of inflammatory markers can be helpful in understanding the inflammatory course. We have now added the absolute concentrations (pg/mL) of the inflammatory markers (Extended Data Figure 4).

* Figure 4d: The results are described as 'tracking GD2+ CAR T-cells in CSF after IV and ICV administration revealed more CAR T-cells detected in CSF at times of peak inflammation, and

higher CAR T-cells in CSF after ICV administration than after IV administration for DIPG Subject 2 (Figure 4d). DIPG subject 3 similarly received both an IV and ICV CAR-T infusion: what did the comparison of CAR-T as detected in CSF for this patient show?

We thank the Referee for this perceptive question. Following IV CAR administration, we did not identify CAR T cells in the CSF of DIPG subject 3 by scRNA-seq at the CSF collection timepoints (Day 7 and 28). Following ICV CAR administration, DIPG subject 3 did not require CSF drainage during anticipated peak inflammation and given her young age, elective CSF collection was more limited. Therefore, this patient's only post-ICV timepoint for CSF collection was Day 14, at which time we did not find enough cells in CSF to conduct scRNAseq.

* Extended data Figure 1b - please include normal cortex comparator similar to as shown in Figure 1c

Thank you for pointing out these necessary comparators. We have now included cortex staining of both CD4 and CD163 in Extended Data Figure 1c and 1f, and shown below. The legend was updated to state "Cortex CD4+ staining depicts rare leptomeningeal vascular CD4+ cells and serves as an internal positive control." and "Cortex CD163 staining demonstrates microglial cells in their "resting" perivascular location. No activated microglia or macrophages are present." (Lines 649-651 and 659-661).

LEFT (Extended data figure 1c): Post-mortem pons tumor tissue of DIPG Subject 1 show evidence of CD4+ and CD8+ T-cells. Unaffected cortex CD4+ staining depicts rare leptomeningeal vascular CD4+ cells and serves as an internal positive control.

RIGHT (Extended data figure 1f): CD163+ myeloid cells in tumor tissue of DIPG Subject 1. Unaffected cortex CD163 immunostaining demonstrates microglial cells in their "resting" perivascular location. No activated microglia or macrophages are evident in the normal cortex of DIPG Subject 1.

* Extended data figure 3d: please explain what Sup_1, Sup_10 etc refer to.

Thank you for pointing this out. We have revised Extended Data Figure 3 to clearly describe the data points relevant to ctDNA assay development, as shown below.

* Extended Table 1: please include time off treatment when treatment on this phase 1 clinical study started

We thank the Referee for their comment and have included a column describing “Time from prior treatment to CAR infusion” into Extended Table 1 (Lines 772-777).

Referee #2 (Remarks to the Author):

Title: "GD2-CAR T-cell therapy for H3K27M-mutated diffuse midline gliomas" by Professor Monje and co-authors (manuscript number 2021-07-11811)

Summary: Majzner, Ramakrishna et al. describe findings from their phase 1 clinical trial (NCT04196413) to treat H3K27M-mutated diffuse midline gliomas and demonstrate clinical improvement following CAR T cell treatment in three of four patients with manageable toxicity profiles. In addition, they perform multiple lines of molecular profiling and provide valuable insights for heterogeneity in response. Overall, the authors demonstrate very promising tolerability and response results for the use of CAR T cells targeting the disialoganglioside GD2 for H3K27M diffuse midline gliomas.

Comments: This manuscript is well-written, clear and is beautifully illustrated. Dr. Monje is a world leader in this field and this exceptional study (in the true spirit of a physician scientist) is based on a previous landmark preclinical paper in H3K27M models. The current study is superlative and is a major advance in not only understanding the biology of these tumors but also provides a promising treatment for this fatal childhood cancer. They provide great detail in their characterization of toxicities and their responses, which will serve as valuable data for future immune-driven approaches to treating brainstem-associated and other tumors. This exceptional paper will be of great interest to the pediatric brain tumor community, and the correlative supporting data will serve as a valuable example of translational research for rare tumors. The following minor comments are noted to improve the clarity and main message of this paper:

We thank the Referee for their detailed review of this manuscript and their thoughtful and supportive comments. We have addressed the Referee comments below in point-by-point responses.

Minor comments:

1) The single-cell RNA-seq data provide incredible, valuable insight into potential predictive factors about engineered CAR T cells and their likelihood of producing a clinical response. As illustrated in Ext. Data Table 2, there are considerable differences in cell manufacturing for DIPG 1 compared to the other tumors. Do the authors have any metrics on the initial autologous peripheral blood mononuclear cell populations that may shed light on the differences in the resulting CAR T % and CD4+/CD8+ breakdown?

We thank the Referee for these gracious comments. In response to this important question, we have added data on the initial CD4/CD8 enriched apheresis sample that was used for CAR T-cell manufacturing in Extended Data Table 2. The CD4/CD8 ratio in the apheresis product does not explain differences in resultant CAR T % or CD4/CD8 breakdown. Future studies will be required to understand these differences and potentially address them to enhance CAR T-cell functionality in patients.

2) The authors primarily compare DIPG 1 and DIPG 2 populations from either the starting product or from CSF myeloid cells to identify pathways that may explain the differences in clinical response. Do any of the findings highlighted in Fig. 4e and f persist in comparisons to the other two tumors? If not, are there other changes that may explain the differences in clinical response?

We have added comparisons of T-cell products for A) DIPG 1 to Spinal DMG 1 and B) DIPG 1 to DIPG 3 to the supplemental figures (Extended Data Figure 6b and shown below).

DIPG 1 to Spinal DMG 1

DIPG 1 to DIPG 3

Extended Data Figure 6b: Gene ontology term analysis of the most significantly activated pathways in DIPG Subject 1 CAR+ product, relative to Spinal DMG 1 or DIPG 3 CAR+ product, prior to CAR T-cell administration. The y-axis shows the enhanced gene ontology terms, while the x-axis (“GeneRatio”) corresponds to the overlap between the up-regulated genes and genes associated with the given gene ontology terms in the dataset. Color of the circle corresponds to the p-value significance of the pathway enrichment, relative to all other genes in the dataset. Size of the circle corresponds to the relative number of matching up-regulated genes in the given gene ontology term.

We also compared the myeloid compartment from DIPG Subject 1 to the other patients/subjects. This demonstrated signatures of increased neutrophil degranulation and interleukin signaling in DIPG Subject 1 at peak timepoints, suggesting that these myeloid pathways may be implicated in the poor response observed in DIPG Subject 1.

CSF myeloid cells DIPG1 vs DIPG2

CSF myeloid cells DIPG1 vs DIPG 3

Extended Data Figure 8: Gene ontology term analysis of myeloid cells from CSF following CAR T-cell treatment. Gene ontology term analysis demonstrates the most significantly activated pathways in the myeloid fraction of DIPG Subject 1 compared to DIPG Subject 2, DIPG Subject 3, and Spinal DMG Patient 1. The y-axis shows the enhanced gene ontology terms, while the x-axis (“GeneRatio”) corresponds to the overlap between the up-regulated genes and genes associated with the given gene ontology term in the dataset. Color of the circle corresponds to the p-value significance of the pathway enrichment, relative to all other genes in the dataset. Size of the circle corresponds to the relative number of matching up-regulated genes in the given gene ontology term. Only myeloid cells in CSF are included in this analysis.

3) Considering the differences in the starting population of DIPG 1 CAR T cells compared to the other tumors (Ext. Data Table 2), can the authors provide information (if available) on the percentage of GD2-CAR T cells that make up the CSF population for later time points in DIPG 1 as done for DIPG 2 and Spinal DMG. Additionally, characterization of the CD4/8 ratio of these CSF GD-2 CAR T cells in the various patients could shed light on potential relationships with clinical response.

We thank the referee for this interesting point. Unfortunately, CAR T-cells were not identified by scRNA-seq in CSF at post-IV treatment timepoints for DIPG Subject 1 or DIPG Subject 3, and DIPG Subject 3 had no cells present in CSF obtained following ICV CAR administration. While our patient numbers remain limited, both the patient with best IV-treated response (DIPG Subject 2) and the two ICV-treated patients (DIPG Subject 2 and Spinal DMG Patient 1) demonstrated a trend to increased proportion of CD8 T cells at later timepoints (either Day 14-28 after IV administration or Day 12-14 after ICV administration). These data are now included in the manuscript (Extended Data Figure 6b-c).

Extended Data Figure 6c: Pie charts represent CD4:CD8 ratios from patients at different timepoints following CAR T-cell administration.

4) The authors describe a population of cells in Ext. Data Fig5d and e that is unique to DIPG 1. Does this population remain independent if myeloid cells from the other tumors are incorporated? What genes drive this cluster?

We thank the Referee for highlighting this population and for the helpful question. In response, we repeated the analysis using all four patient peak timepoint samples, and determined that this population is not unique to DIPG Subject 1.

We have now conducted in-depth analysis of the myeloid populations, identifying an interferon-gamma response myeloid population derived primarily from the ICV early expansion timepoints. Conversely, IV treatment and late CSF timepoints exhibited a strong immune suppressive profile, which aligned with “disease-associated microglia” (DAM) cell state, myeloid-derived suppressor cell (MDSC) state, and “axon tract-associated microglia” (ATM) cell state signatures (Figure 6 and shown below).

Figure 6: Single cell transcriptomic analyses identifies distinct myeloid subpopulations

(a) Patient CSF sample scRNAseq dataset was filtered to isolate myeloid cells. Clustering was conducted after data integration by Harmony.

(b) Pie charts represent myeloid cluster proportions at different timepoints. A single UMAP was generated with myeloid cells from the CSF then individually visually represented based on day of the sample, colored by patient. Note alterations in presence of myeloid clusters over time.

(c) Expression signature of “immune activating”, “immune suppressive”, “DAM Stage 1”, “DAM Stage 2”, “MDSC”, and “ATM” cell states and associated representation within clusters based on single-cell expression scores (Z-score).

5) Can the authors provide clarity on the figure legend for Fig 1e. Does “tumor volume” include both spinal and brain disease?

We thank the Referee for their attention to this detail that requires clarification. Figure 1e represents the tumor volume in the primary site (spinal cord) and does not represent the brain disease. We have now clarified this in the figure legend.

6) It is very interesting that the DIPG 1 GD2 CAR DNA levels do not appear to drop as they do in the other patients' samples. Could this be related to the clinical response differences?

We appreciate the Referee's comment and agree that it is of interest that the GD2-CAR DNA levels remain elevated in DIPG Subject 1. Based on our scRNAseq and CSF cytokine data analyses, it appears that there may have been a more immune suppressive environment in DIPG Subject 1, with elevated immune suppressive cytokines and myeloid cells, implying that the CAR T-cells may have been unable to function effectively in the tumor microenvironment. This may explain the poor response despite persistent CAR T-cells. As we accumulate more patients in the trial, we will evaluate to see if this pattern is present in other patients on the trial as well.

Referee #3 (Remarks to the Author):

The authors present a series of four patients who have been treated on a novel and exciting clinical trial in which they were treated with intra-CNS immunotherapy for DIPG.

The plan and execution of the clinical trial are novel, innovative, and likely first in class for the globe. This is an exciting avenue for the future treatment of brain tumors. The documentation of the cases is excellent, and there is an amount of correlative science to accompany the clinical data.

Unfortunately, this is a series of only four patients, and while the preliminary results are interesting, I would suggest that the authors be encouraged to resubmit once they have a larger cohort, as this cohort is too small to draw any conclusions, and will be of limited interest at this point to the broad readership of Nature.

I would be highly in favor of inviting the authors to resubmit when their cohort is larger, and when more of the patients have been observed to endpoint.

We appreciate the positive and supportive comments of this Referee. While this manuscript is limited to the first four patients, we believe that this paper is important to share now as it highlights important lessons for CAR T cell therapy of brainstem tumors, as well as

immunotherapeutic approaches for brain tumors in general. As instructed by the Editor, we have added the caveat that these data represent only our Dose Level 1 experience, with limited patient numbers, stating in the discussion that “the clinical experience reported here is early and the number of subjects treated to date is small...” (Lines 437 – 439). We look forward to continuing this important work to optimize this approach for patients with H3K27M+ DIPG/DMG.

Reviewer Reports on the First Revision:

Referee #1 (Remarks to the Author):

The authors have now included additional information as requested to provide the reader with a complete clinical picture of the patients treated on this study including objective measures of radiological responses.

For all 4 patients measured tumor volumes are now presented. The only missing time point is d+75 for spinal DMG patient - please include this for completeness in figure 2g.

A clear explanation is provided as to why a comparison of detection of GD2-CART cells in blood and CSF as provided for DIPG patient 2 could not be performed for DIPG patient 3. Please include this information in the text or figure legend of figure 4d.

Referee #2 (Remarks to the Author):

The authors have addressed all concerns. This is an important paper in the field.

Referee #3 (Remarks to the Author):

The authors have now answered all of the concerns raised by the reviewers, and the manuscript will make an important addition to the literature.

Author Rebuttals to First Revision:

Referee #1:

The authors have now included additional information as requested to provide the reader with a complete clinical picture of the patients treated on this study including objective measures of radiological responses.

For all 4 patients measured tumor volumes are now presented. The only missing time point is d+75 for spinal DMG patient - please include this for completeness in figure 2g.

We have now included the day +75 timepoint for spinal DMG Patient-1 (now Figure 2f).

A clear explanation is provided as to why a comparison of detection of GD2-CART cells in blood and CSF as provided for DIPG patient 2 could not be performed for DIPG patient 3. Please include this

information in the text or figure legend of figure 4d.

We have now included the following clarifications:

-in the main text, “...*limited CSF samples were obtained from DIPG Subject-3 due to her young age.*”

-in the Figure legend (now Extended Data Figure 6b): “*CSF studies were also limited in DIPG Subject-3: Following ICV GD2-CART administration, DIPG Subject-3 did not require CSF drainage during the period of peak inflammation and given her young age, elective CSF collection was more limited. Therefore, the only post-ICV timepoint for CSF collection from DIPG Subject-3 was Day 14, at which time we did not find enough cells in CSF to conduct scRNAseq.*”

Referee #2 (Remarks to the Author):

The authors have addressed all concerns. This is an important paper in the field.

We thank the Referee for these positive comments.

Referee #3 (Remarks to the Author):

The authors have now answered all of the concerns raised by the reviewers, and the manuscript will make an important addition to the literature.

We are grateful for the positive comments.